# VoxelPrompt: A Vision System for Language-Prompted Medical Image Analysis

## Abstract

We present VoxelPrompt, an end-to-end vision system that tackles composite radiological tasks. Given a user prompt, VoxelPrompt integrates a language model that generates executable code to invoke a novel, jointly-trained vision network. This adaptable network can integrate any number of volumetric (3D) inputs across heterogeneous real-world clinical modalities to segment and characterize diverse anatomy and pathology. Predicted code employs this network to carry out analytical steps to automate practical quantitative pipelines, such as measuring the growth of a tumor across visits, which often require practitioners to painstakingly combine multiple specialized but brittle tools. We evaluate VoxelPrompt using diverse neuroimaging tasks and show that it can delineate hundreds of anatomical and pathology features, measure complex morphological properties, and perform open-language analysis of lesion characteristics. VoxelPrompt performs these objectives with an accuracy similar to that of specialist single-task models for image analysis, while facilitating a broad range of biomedical workflows.

## 1 Introduction

Clinicians and scientists routinely pose complex questions involving specific targets in medical imaging, which extend well beyond simple segmentation or classification tasks. These questions involve multi-step efforts to track the evolution of a particular pathology over many scans, quantify subtle asymmetries of a specific anatomy, or integrate information from multiple acquisitions.

As a detailed example, consider tracking the growth of a specific lesion over time in a patient with multiple abnormalities. After image pre-processing, the first challenge is segmenting *only* the specific lesion of interest. Available tools rarely generalize to diverse, real-world lesion types, and even those that do, offer no way to identify a specific lesion using natural language descriptors (e.g., by anatomical location, size, or intensity). Additionally, current tools do not typically accommodate a flexible number of acquisitions from a scan session. As a result, the user must choose a single suitable scan, develop a custom pipeline to programmatically select the target lesion, repeat the process for later scans, and then compute the desired downstream metrics to track changes.

The example above illustrates a fundamental barrier in integrating AI in real imaging workflows. While existing specialized tools perform well for particular segmentation or classification targets (Billot et al., 2023; Isensee et al., 2025), they involve strict data assumptions and cannot broadly adapt to nuanced, case-specific real-world goals that require individualized analyses. This task-level specialization limits the adoption of AI in radiology, as it forces practitioners with complex radiological questions to manually chain together multiple fragile, task-specific, and often inadequate models and develop extensive post-processing and metric-extraction workflows for each new study.

VoxelPrompt is fundamentally different in functionality and design from existing medical image analysis systems. In VoxelPrompt, we jointly train a language model and vision network from scratch to generate and execute *end-to-end* image analysis workflows. Given a task described in natural language, the language model iteratively predicts a sequence of instructions as executable code. The dynamically evaluated instructions generate spatial features (e.g., segmentations) using the vision network, incorporate natural language responses, and access a library of functions to compute and provide quantitative outputs. Through diverse output modalities, VoxelPrompt can segment and localize user-specified anatomical and pathological regions of interest, calculate measurements that relate multiple scans to one another, and perform biomedical characterization (Fig. 1).

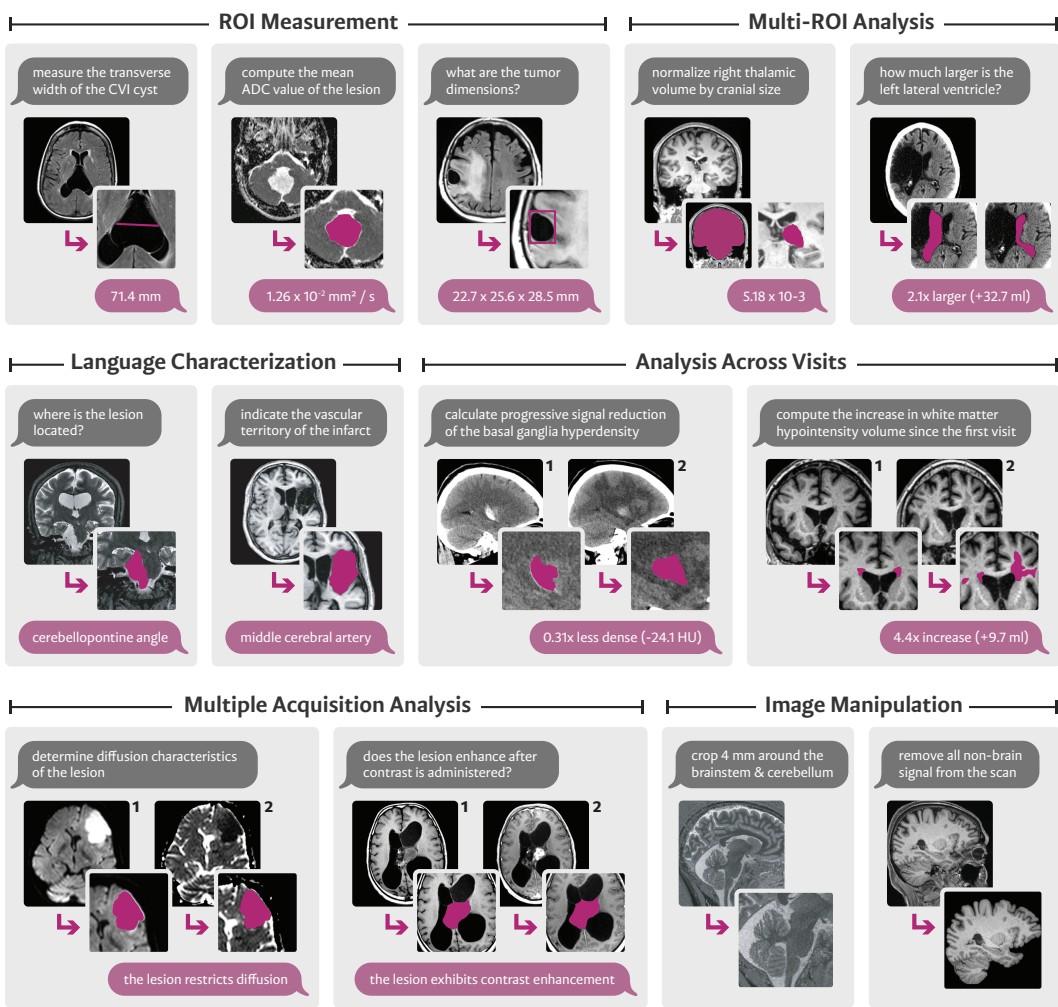

Figure 1: Illustrative examples of VoxelPrompt capabilities, each showing the input prompt (gray) and volumes with VoxelPrompt's predicted annotations and language responses (purple).

We make several technical contributions to realize VoxelPrompt's capabilities across diverse, real-world medical imaging aims. Our convolutional vision network enables fine-grained, language-conditioned visual analysis by integrating jointly-trained language model embeddings and generating spatial segmentation maps and encodings for downstream image characterization. To support multi-acquisition and longitudinal analysis (Reuter et al., 2012), the vision network uses attention to interact volumetric features across an input set of scans of any length. Further, unlike existing methods, which are tied to fixed image geometries (Zhang et al., 2024), VoxelPrompt operates on variable-sized inputs at native voxel resolution, which yields substantial efficiency improvements, thereby enabling the joint training of vision and language networks on large 3D volumes using minimal GPU resources. Lastly, we facilitate robustness to acquisition type as well as anatomical and pathological variation by creating and training on a large neurological dataset combining public datasets, new annotations of unlabeled pathological volumes, and simulated lesions.

We focus on brain imaging and show that VoxelPrompt enables end-to-end analysis on nuanced and diverse tasks covering a wide range of MRI and CT acquisitions, anatomies, and diseases. We show that a single VoxelPrompt model captures, and often exceeds, the individual accuracy and capabilities of many single-task specialist neuroimaging baselines, while retaining unique flexibility across diverse tasks, regions of interest, and acquisition properties. These results highlight VoxelPrompt's promise as a foundation for tackling diverse and complex radiology workflows.

## 2 RELATED WORK

**Brain Region Analysis.** Widely-used neuroimage analysis pipelines typically delineate regions and quantify their size, shape, composition, and change over time (Fischl, 2012; Jenkinson et al., 2012). Modern approaches train networks to segment anatomical and pathological structures, including cerebral subregions (Billot et al., 2023; Henschel et al., 2020), vessels (Hilbert et al., 2020; Livne et al., 2019), and lesions (Hssayeni et al., 2020; Liu et al., 2021). While performant, these networks generally work for fixed segmentation targets and require significant human involvement for analyzing data and deriving downstream ROI measures. VoxelPrompt aims to match or outperform these methods in segmentation accuracy, while tackling a wider range of targets, enabling flexible specification of target regions, and facilitating end-to-end workflows.

**Learning Across Medical Imaging Tasks.** Recent medical imaging methods aim to improve performance by exploiting shared representations across diverse segmentation, classification, registration, and statistical modeling objectives in a single framework (Elmahdy et al., 2021; Graham et al., 2023; Tellez et al., 2020; Liu et al., 2025; Czolbe & Dalca, 2023). Broad, segmentation-focused tools, like interactive or in-context segmentation models, can adapt to specific biomedical targets, prompted by partial image annotations (Cheng et al., 2023; Luo et al., 2021; Ma et al., 2024; Wong et al., 2023) or example image-segmentation pairs (Min et al., 2021; Xie et al., 2021; Butoi et al., 2023; Ouyang et al., 2022; Rakic et al., 2024; Roy et al., 2020). However, these multi-task models do not aim to address a complete analytical pipeline and can require finetuning in real scenarios. In contrast, VoxelPrompt integrates supervision from many tasks to create computational workflows, where multiple components interact to carry out requested analyses.

**Medical Vision-Language Models.** Vision-language models (VLMs) trained on large-scale biomedical image-caption datasets (Johnson et al., 2019; Lin et al., 2023; Zhang et al., 2023a) can facilitate biomedical visual question-answering (Chen et al., 2023a;b; Zhang et al., 2023a;b) and clinical report generation (Bannur et al., 2024; Wang et al., 2023c;b). However, current biomedical VLMs remain largely limited to narrow-domain, text generation tasks, and do not capture the quantitative metrics required in real-world clinical imaging workflows. In contrast to current vision-language models that produce text outputs in a black-box manner, VoxelPrompt explicitly produces code for all relevant intermediate outputs and a traceable sequence of operations. This provides analytical transparency for high-stakes applications. Also, unlike existing models, the VoxelPrompt operations involve explicit vision operations to compute and present images depicting the essential intermediate features. Finally, except for a few recent works (Chen et al., 2023a;b; Liu et al., 2023; Wu et al., 2025; Zhou et al., 2024), most models are trained exclusively on two-dimensional image slices, often X-rays, making them inappropriate for MR and CT imaging. VoxelPrompt is instead trained directly at native acquisition resolution, enabling it to process 3D volumes.

**Language Models as Agents.** Recent work extends large language models beyond plain text prediction into agents capable of planning and executing actions for computational tasks. Often, these generate code (Gupta & Kembhavi, 2023; Ke et al., 2025) that call external APIs for mathematical computation (Ruan et al., 2023; Gou et al., 2023), image analysis (Li et al., 2024; Subramanian et al., 2023; Surís et al., 2023; Yang et al., 2023), scientific discovery (Bran et al., 2023; Boiko et al., 2023), and more. Adaptive, feedback-driven agents address complex and dynamic problems by iteratively planning, executing, and interpreting intermediate outcomes rather than predicting entire action sequences at once (Huang et al., 2022; Rana et al., 2023; Wang et al., 2023d;a; Yao et al., 2022; Zhu et al., 2023). Building on this idea, VoxelPrompt trains a language model that interacts with a library of processing functions. However, since specialized neuroimaging tools do not easily adapt for diverse, context-specific pathology targets, we jointly train an adaptable vision model from scratch for flexible, language-controlled analysis.

## 3 METHODS

### 3.1 MODELING DETAILS

**Overview.** In response to a text prompt, VoxelPrompt processes a set of $n$ subject-specific image volumes $\mathcal{V} = \{v_i\}_{i=1}^{n}$. A volume consists of a feature tensor in $\mathbb{R}^{c \times w \times h \times d}$ (with $c$ channels and $w, h, d$ spatial dimensions) and a world-coordinate matrix in $\mathbb{R}^{4 \times 4}$ specifying voxel spacing and position.

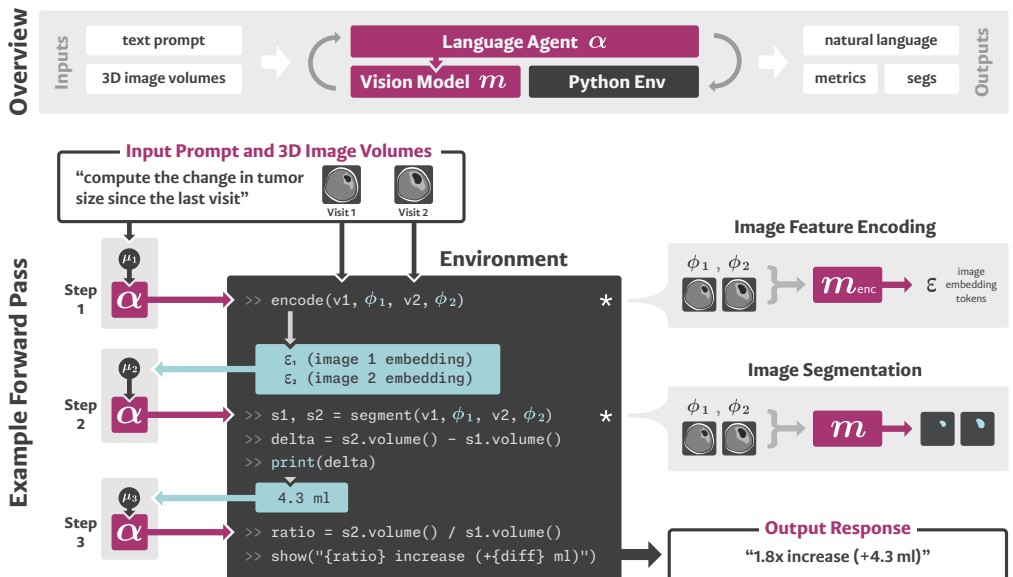

Figure 2: **Top:** VoxelPrompt takes a text prompt and volumes as input to a trainable language model $\alpha$, which iteratively produces executable code in a Python environment, which controls a jointly-trained vision model $m$. **Bottom:** To solve an example language-prompted task, the model $\alpha$ interprets execution outcomes $z$ (blue) to guide subsequent instruction prediction across multiple steps. To perform vision operations, such as volume encoding or generation, $\alpha$ employs $m$, which is manipulated by image-specific latent instruction embeddings $\phi$.

A language model $\alpha$ translates the prompt into code, executed in a persistent Python environment, to invoke actions involving mathematical computation, morphological operations, and interface interaction. A core function set runs a jointly trained vision network directed by the language model to perform vision operations. Figure 2 summarizes this framework and outlines an example use-case. We provide low-level implementation details and a table of notation in Appendix A, and we attach our library code of morphological functions available to $\alpha$ as supplementary material.

**Language Model.** The language model $\alpha$ predicts code iteratively, with each step building on outcomes of prior actions. At step $i$, it generates executable code $k_i = \alpha(\mu_i)$ based on a state representation $\mu_i \in \mathbb{R}^{\ell \times b}$ with sequence length $\ell$ and token embedding dimension $b$. The code executes within the python environment and resulting outputs are embedded into token representation $z_i$ and incorporated into the next state $\mu_{i+1} = \mu_i \parallel z_i$, where $\parallel$ denotes sequence concatenation. The initial state $\mu_1$ encodes the prompt and acquisition date metadata for each volume $v \in \mathcal{V}$. This loop of code generation, execution, and feedback continues until a predicted stopping token signals completion.

**Vision Model.** Functions in the python environment can invoke a convolutional vision network $m(\cdot)$ to produce segmentations or image embeddings interpretable by $\alpha$. To modulate vision operations, the language model generates embedding tokens $\phi \in \mathbb{R}^{n \times b}$ corresponding to the $n$ input volumes. We implement $m$ as a UNet-style architecture with a downsampling encoder ($m_{\text{enc}}$) and an upsampling decoder. The vision network can generate a set of $n$ segmentation volumes $\mathcal{W} = m(\mathcal{V}, \phi)$, with each segmentation corresponding to the respective input in $\mathcal{V}$. For outputs associated with multiple scans within a single scan visit, we max-pool the results into one segmentation map. The vision network can also generate image embeddings $\varepsilon \in \mathbb{R}^{n \times b}$, using only the encoder $\varepsilon = m_{\text{enc}}(\mathcal{V}, \phi)$, which are concatenated into $z_i$ as language model feedback.

**Native Geometry Processing.** VoxelPrompt operates on input $\mathcal{V} = \{v_i\}_{i=1}^n$, where each $v_i$ has variable spatial dimensions, voxel spacing, and physical position in world coordinates. Prior approaches that resample all inputs in a set to a common high resolution require substantial memory. Instead, we propagate each volume through the vision network $m$ at its native spatial geometry, applying convolutional operations independently. We maintain world-coordinate matrices alongside intermediate activations of each input, and update them at each encoder layer to reflect the voxel

scale. During downsampling, we only pool along a certain axis if the anisotropy ratio is lower than the pooling factor. During decoding, target upsampled resolutions are inferred from the respective geometry of the corresponding skip-connection feature maps.

**Cross Volume Conditioning.** Each volume in an input set is processed through the vision network in parallel, with weights shared across the set. After each convolutional block, we interact all features of the set via cross-attention across volumes at matching spatial indices, enabling features at corresponding spatial locations to exchange information across inputs, conditioned by the language instruction $\phi$. To enable interaction across heterogeneous geometries, the intermediate activation maps are first trilinearly resampled to a shared reference grid using the minimum voxel spacing across volumes. We stack these resampled maps to obtain, for each voxel, feature tensors in $\mathbb{R}^{n \times c}$, where $c$ is the number of intermediate activation channels. For each voxel, we concatenate the condition $\phi$ along the channel dimension and process the result with a single-layer MLP mapping $\mathbb{R}^{n \times c + d} \rightarrow \mathbb{R}^{n \times c}$. We then apply a transformer layer to compute projected, voxel-wise cross-attention across the $n$ volumes at matched spatial indices, and resample the resulting maps back to their original geometries. We apply this operation at all convolutional levels, except the highest resolution.

**Supervised Training.** We jointly train $\alpha$ and $m$ from scratch on a curated task set $\mathcal{T}$ (Section 3.2). Each task $\tau \in \mathcal{T}$ is paired with target (ground-truth) code $k^*$ that carries out the task objective, as illustrated in Figure 2. At each training step, we sample $\tau \sim \mathcal{T}$, generate a prompt, and sample input volumes $\mathcal{V}$ with ground-truth outputs $\mathcal{W}^*$. The training loss is $\mathcal{L}_{ce}\big(P(k), k^*\big) + \lambda \sum_{j=1}^{|\mathcal{W}|} \mathcal{L}_{seg}\big(\mathcal{W}_j, \mathcal{W}_j^*\big)$, where $P(k)$ is the predicted token distribution over the language model vocabulary, $\mathcal{W}$ are segmentations produced by $m$ when executing $k^*$, $\mathcal{L}_{ce}$ computes cross-entropy over tokens, and $\mathcal{L}_{seg}$ compares predicted and target volumes using a soft Dice loss.

### 3.2 TRAINING TASKS AND DATA DESIGN

We curate a dataset $\mathcal{T}$ of clinically oriented brain imaging tasks, spanning a wide range of image acquisitions, segmentation protocols, and annotation types. We use this dataset to both train and evaluate VoxelPrompt in the joint prediction of analytical instructions, spatial delineations, and natural language descriptions. We categorize tasks as quantitative processing or qualitative characterization. Quantitative processing tasks involve segmenting and computing measures such as the spatial dimensions, volumetric change, or signal statistics of a specific region of interest (ROI). Qualitative characterization requires language descriptions of pathology, such as a lesion's anatomical location or signal properties.

**Task Construction.** For each task $\tau \in \mathcal{T}$, we develop a strategy to sample the relevant input prompt, images, ground-truth segmentations, processing code, and natural language outputs needed to solve that task. Appendix B.1 enumerates the definition of all task classes, including their objectives, example prompts, expected inputs and outputs (metrics or text).

During training, we construct a single supervised example per model pass. We first sample a task $\tau$, then sample all associated parameters, including the ROIs, image–segmentation pairs, task parameters, and an input prompt. We determine the target output code $k^*$ deterministically from a manually defined template for each task type. For quantitative ROI-processing tasks, $k^*$ calls encoding and segmentation routines to process the volumes and then applies morphometric functions to the resulting segmentations to compute the task-associated measures. For qualitative characterization tasks, $k^*$ produces a natural language response using a task-specific template that maps predefined pathology annotations to a structured description. The resulting ground-truth text is consistent – for example, if a sampled tumor is annotated as "parietal" and "left", the target location description is always formatted as "left parietal lobe."

**Training Prompt Synthesis.** We synthesize a diverse set of prompts for training. For each task $\tau$, we manually construct a set of $10 - 30$ prompt templates $\mathcal{P}_\tau$, each defined using semantic slots that correspond to interchangeable terminology and phrasing relevant to the task (Appendix Figure 6). Each slot $p$ is associated with a curated synonym set $\mathcal{C}_p$, and generating a prompt involves: (1) sampling a template from $\mathcal{P}_\tau$ and (2) sampling a candidate substitution from $\mathcal{C}_p$ for each slot. Some task-specific templates use recursion, where a slot may contain additional slots. For instance, nested

noun-phrase patterns such as *"tumor segmentation"* may optionally receive prefixes such as *"derive a ..."* or *"compute the ..."*, corresponding to typical user instructions.

We generate target ROI descriptions with variable degrees of specificity. For each anatomical region and abnormality class in our dataset, we manually define a list of alternative terminologies. For disease classes, we construct a hierarchical tree to group abnormalities based on their ontology (e.g., glioma → neoplasm → mass). To sample an ROI description, we first randomly select a class-specific terminology. If an abnormality ROI is the only abnormal finding tagged in the associated image, we optionally sample parent terms from the hierarchy. For example, a glioma could instead be randomly described as a neoplasm, mass, or generic lesion. As discussed in the following section, disease instances in our dataset include metadata that defines properties such as location or signal. We use each metadata tag as an adjective slot inserted randomly as a prefix or suffix around the selected ROI terminology, for instance as *"left frontal mass"* or *"mass in the frontal lobe"*. If multiple abnormalities are present in a subject, we insert all metadata in the ROI description to avoid ambiguity, otherwise we randomly drop metadata tags from the description to produce sparser phrasing corresponding to coarse-grained prompts.

**Training Images and Segmentations.** We assemble and annotate a collection of 6,925 3D brain MRI and CT scans from 15 public datasets, comprising 185 bilateral anatomical structures and 14 pathology classes, focusing on a breadth of imaging types, regions of interest, and tasks. The MRI sequences span T1w, T2w, FLAIR, PD, GRE, and DWI with various scan resolutions. The subjects are split into 4,852 training, 213 validation, and 1,860 test volumes. Anatomical segmentations are derived from established pipelines (Fischl, 2012; Greve et al., 2021; Hoopes et al., 2022), atlas annotations (Adil et al., 2021; Pauli et al., 2018), manual corrections, and manual labeling of additional structures in a small set of images, yielding high-quality whole-brain labels across multiple cohorts.

To capture diverse pathologies, we integrate expert-annotated lesions from BraTS, ISLES, ATLAS, and WMH (Baid et al., 2021; Hernandez Petzsche et al., 2022; Liew et al., 2022; Kuijf et al., 2019), covering gliomas, edema, infarcts, and white matter hyperintensities. We further compile rare cases from *Radiopaedia* and manually delineate infarcts, arachnoid and epidermoid cysts, papillomas, and many others. These new annotations also include sub-components like edema, enhancing tissue, and heterogeneous intra-lesion features. Finally, we augment the dataset with a conditional synthesis procedure that generates diverse lesions in healthy brains, broadening the distribution of pathological presentations (Appendix B.6). To support analysis of lesion characteristics, we annotate each lesion with its anatomical location, intensity profile, size, and position relative to surrounding structures, and, when applicable, indicators of diffusion restriction or post-contrast enhancement.

## 4 EXPERIMENTS

VoxelPrompt addresses composite radiological workflows rather than a single fixed task. As there exists no benchmark dataset for workflows, its evaluation requires a diverse set of complementary experiments focused on its individual capabilities. We first evaluate VoxelPrompt's ability to execute accurate end-to-end brain analyses across several representative practitioner use-cases. We then present analyses and ablations of modeling decisions. We provide further experimental details in Appendix C and include additional results for disease characterization performance in Appendix D.

### 4.1 BRAIN IMAGE ANALYSIS

***Ad hoc* Neuroimaging Workflow Generation.** Figure 1 shows that a single VoxelPrompt model can execute a wide range of workflows on held out test data, including localizing brain anatomy and pathology regions, extracting intensity metrics and morphology measures within user-specified ROIs, and masking or cropping tissues for focused visualization. The model can compute and compare metrics across ROIs, such as hippocampal asymmetry, normalized subcortical volumes, and acute versus chronic hemorrhage components, as well as track longitudinal changes such as tumor size across scans. By integrating multiple acquisitions, VoxelPrompt can further characterize lesion locations and tissue properties, such as diffusion restriction or post-contrast enhancement. Figure 3A shows that VoxelPrompt facilitates fine-grained specificity by supporting flexible, language-guided analysis, such as isolation or differentiation of lesions in multifocal disease based on signal intensity, size, relative position, or anatomical context (e.g., hemisphere, lobe, etc.). Examples in Figure 1

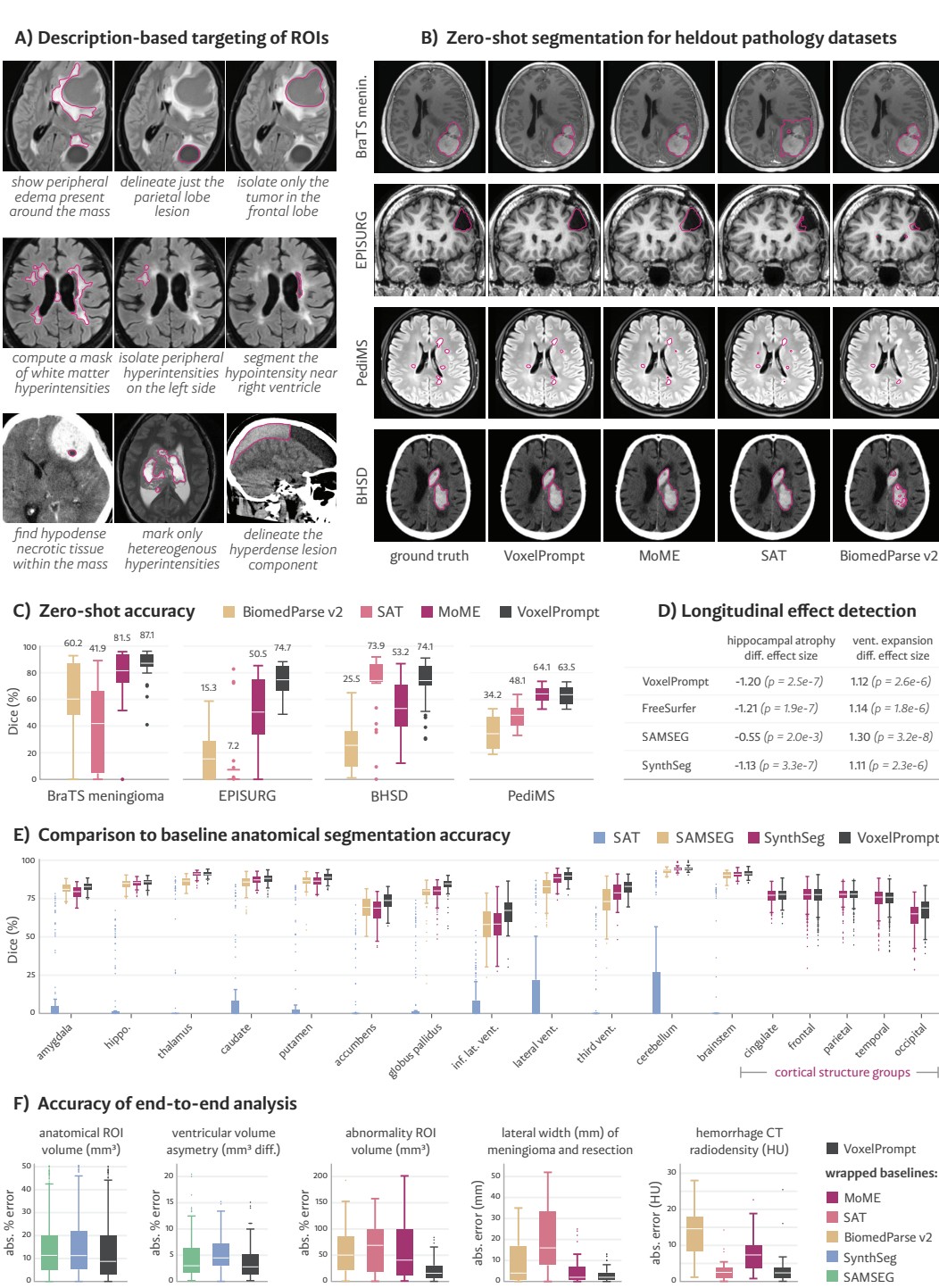

Figure 3: **VoxelPrompt performance. (A)** Free-form text prompts, shown below each image, guide VoxelPrompt to perform targeted analysis and delineation of nuanced, context-specific image regions, even in scans with multiple lesions. **(B, C)** On unseen datasets with diverse brain abnormalities, VoxelPrompt is the only method achieving consistently high-quality results both qualitatively and quantitatively. **(D)** Compared to multiple longitudinal-specific baselines, VoxelPrompt achieves comparable effect sizes in distinguishing Alzheimer's disease from controls. **(E)** VoxelPrompt outperforms state-of-the-art networks and optimization-based methods for whole brain segmentation. **(F)** VoxelPrompt automatically generates downstream quantitative measures with comparable or better accuracy than baseline methods that require custom integration with post-processing code.

reflect common practitioner use-cases, but many are qualitative as no benchmark dataset currently exists to evaluate free-form workflow generation outcomes for complex pathology analyses.

**Text-prompted Zero-shot Lesion Segmentation.** Zero-shot brain lesion segmentation enables medical practitioners and researchers to rapidly localize and quantify pathologies without requiring a disease-specific model. We evaluate VoxelPrompt's segmentation capabilities on four entirely unseen abnormality datasets using a dataset-specific prompt: "segment the $\langle ROI \rangle$," where $\langle ROI \rangle$ is the target lesion type or informative description. We benchmark our approach against multi-dataset foundation models that include brain pathology segmentation as training tasks. These include volumetric BiomedParse v2 (Zhao et al., 2024; 2025a) and SAT (Zhao et al., 2025b), both of which use text prompts to target abnormalities. Most other existing text-prompted vision models, or vision–language models (VLMs), are limited to question-answering tasks, and do not perform quantitative analyses such as segmentation, thereby precluding them as baselines. We also use MoME (Zhang et al., 2024), a recent generalizable brain abnormality segmentation model. Our evaluation suite spans diverse targets: 30 meningiomas from BraTS-MEN (LaBella et al., 2024), 9 pediatric MRIs of multiple sclerosis (MS) lesions from PediMS (Popa et al., 2025), 35 resection cavities from EPISURG (Pérez-García et al., 2020), and 36 hemorrhages from BHSD (Wu et al., 2023). With the exception of meningioma, VoxelPrompt has not been trained on resection, MS lesion, or hemorrhage abnormality classes, allowing for zero-shot performance assessment.

Figures 3B and C demonstrate that VoxelPrompt is the only method that achieves consistently high performance across all lesion target types, and on average achieves the highest Dice score. We find that no baseline achieves generalization across abnormalities, and the second-best method varies from dataset to dataset. Quantitatively, VoxelPrompt achieves a mean 12.53 Dice points higher than MoME, the overall second-best method. Appendix Figure 8 shows per-subject performances.

**Whole Brain Anatomical Segmentation.** We evaluate VoxelPrompt's ability to segment diverse neuroanatomical targets. Since many brain structures are bilateral, we prompt VoxelPrompt to "segment the left and right $\langle ROI \rangle$" to generate joint segmentations, where applicable. We compare against SynthSeg v2 (Billot et al., 2023), the widely-used state-of-the-art model, as well as SAMSEG (Cerri et al., 2023), a probabilistic segmentation framework, and SAT (Zhao et al., 2025b), a vision-language segmentation model. SAMSEG and SAT support a smaller set of neuroanatomical labels than SynthSeg, so these methods are evaluated on a shared subset of 22 overlapping ROIs. SynthSeg is additionally evaluated on 45 labels.

Figure 3E shows that VoxelPrompt significantly outperforms SynthSeg ($p < 0.05$) on 23/45 ROIs, with a mean Dice improvement of $+1.1 \pm 2.3\%$ over all structures. On the shared ROI subset, VoxelPrompt outperforms SAMSEG and SAT by $5.1 \pm 6.1\%$ and $70.8 \pm 25.7\%$ Dice, respectively. While VoxelPrompt achieves a modest improvement, we emphasize that our main goal is not to widely outperform established tools for segmentation sub-tasks, but rather to provide reliable anatomical segmentations while retaining a broad range of flexible text-prompted targets.

**Longitudinal Analyses.** Figure 1 shows qualitatively that VoxelPrompt performs volumetric analyses across time for various pathologies. Here, we quantify performance for longitudinal analyses of *anatomical* structures, a core component of large neuroimaging studies. We use 100 subjects from ADNI (Weber et al., 2021) with MRI sessions separated by two years, evenly split between healthy controls and individuals with Alzheimer's disease (AD). We assess VoxelPrompt's off-the-shelf ability to measure the change in AD-affected structures over time, and use that to distinguish controls from AD subjects. Specifically, we compare effect sizes against longitudinal FreeSurfer (Reuter et al., 2012), the widely used standard for multi-session analysis, as well as SAMSEG-longitudinal (Cerri et al., 2023) and SynthSeg. As demonstrated in Figure 3D, VoxelPrompt detects well-established longitudinal AD effects, such as hippocampal atrophy and ventricular expansion, with effect sizes comparable to these baselines specialized for longitudinal processing and whole-brain anatomical segmentation.

**End-to-End Performance.** We now evaluate VoxelPrompt's performance on quantitative, multi-step workflows. Since no existing system performs these analyses natively and no benchmark dataset exists for such tasks, we construct post-hoc baselines. For each task, we write code to integrate existing segmentation frameworks into the multi-step analysis, reflecting typical practitioner usage.

We evaluate a subset of task categories that include measurements of (1) volumes of anatomical regions and pathologies, (2) ventricular asymmetry, (3) resection cavity and meningioma dimen-

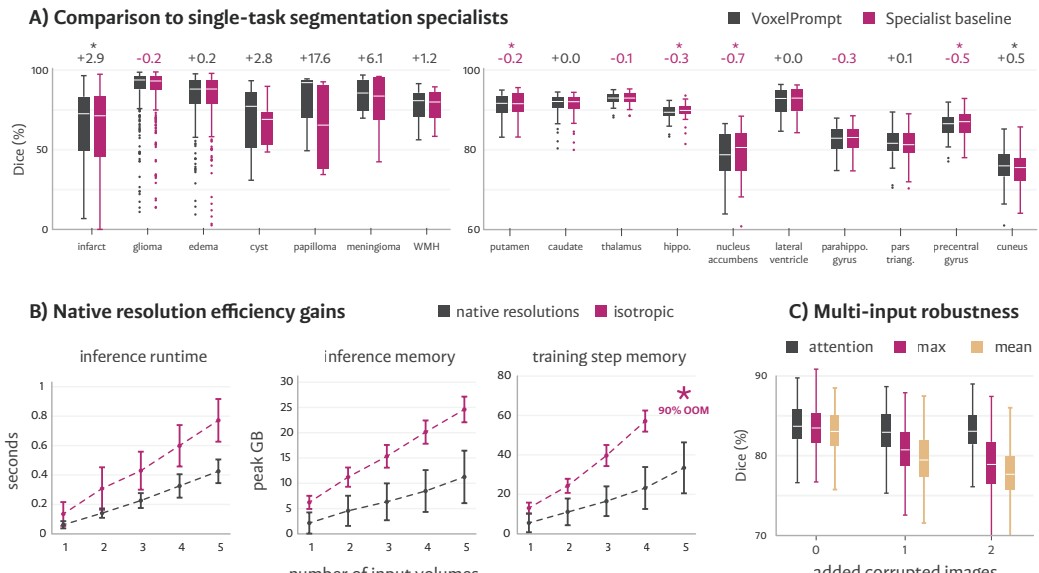

Figure 4: **Ablations and analyses**. **(A)** A single VoxelPrompt model trained jointly on all tasks matches or exceeds task-specific models for both lesions (left) and anatomy (right). Asterisks indicate statistically significant differences. **(B)** Our proposed native-resolution convolutions are more efficient in runtime and memory than isotropic resampling. **(C)** Our attention mechanism for multi-input volume interaction is more robust to image corruptions compared to max and mean reductions.

sions, and (4) mean CT density of hemorrhage lesions. For each task, we compute the absolute error of the quantitative output of VoxelPrompt, as well as the manual wrappers around existing baselines, relative to the ground-truth value. The prompts used for VoxelPrompt in this evaluation are provided in Appendix C.3. As shown in Figure 3F, VoxelPrompt matches or exceeds baseline performance across each end-to-end task, demonstrating that VoxelPrompt can accurately perform end-to-end, multi-step tasks as well as custom pipelines designed specifically for these tasks.

## 4.2 ABLATIONS AND ANALYSES

**Multi-Task Training.** We ablate multi-task training by testing whether a single VoxelPrompt model jointly trained across tasks can match the performance of task-specific specialist networks. These specialists use the same vision backbone as VoxelPrompt (without language conditioning) to build single-label segmentation models for a subset of ROIs. Each specialist is optimized independently, using only its corresponding task labels, with a soft Dice loss. Since optimizing a specialized baseline for each ROI in our training dataset is computationally prohibitive, we select a subset of 10 anatomical and 7 pathology targets spanning diverse shapes and locations. In total, the resulting evaluation subset encompasses 638 held-out subjects.

Figure 4A shows that VoxelPrompt performance is on par with ($p > 0.05$) or exceeds ($p < 0.05$) the performance of $13/17$ single-task specialists. The mean Dice difference relative to the specialists is $+4.3 \pm 5.7\%$ for pathology targets and $-0.1 \pm 0.3\%$ for anatomical structures. This shows that multi-task training in VoxelPrompt rivals specialist models, while offering substantial improvement for brain abnormality segmentation, especially for variable lesions and limited data.

**Native Resolution Efficiency.** We evaluate the efficiency gains of the proposed native-resolution vision network by comparing the processing of images at their native resolution to the standard approach of conforming all inputs to a 1 mm$^3$ isotropic geometry (Billot et al., 2023). For both approaches, we measure inference runtime, peak GPU memory during inference, and training memory incurred during the backward pass of a single optimization step using the VoxelPrompt model. Image geometries are drawn from a distribution reflecting those encountered in a single clinical scan session (Appendix C.4), and results are averaged over 500 samples. To assess scalability, we repeat the experiment with increasing numbers of input images per sample.

Figure 4B shows that, averaged across all numbers of input volumes, native-resolution processing achieves a $2\times$ reduction in inference runtime and a $2.4\times$ memory reduction compared to isotropic resolution conformation. Isotropic resampling in training incurs $2.2\times$ higher memory costs, rendering it challenging to train a multi-modal model that accepts only isotropic inputs on standard hardware. For example, with 5 input volumes, frequently present in a longitudinal MRI series, isotropic inputs cause out-of-memory errors on $90\%$ of sampled batches on an 80 GB GPU.

**Mechanisms of Stream Interaction.** We compare our attention-based interaction module with existing non-parametric alternatives (Butoi et al., 2023) that interact features across image inputs using mean or max reductions. To isolate the effect of interaction, we train vision-only models using three interaction mechanisms: attention, mean-pooling, and max-pooling (Appendix C.6). We train models using synthetic multi-contrast brain images generated from 360 MRIs using a domain randomization pipeline (Gopinath et al., 2024). Synthetic inputs enable us to explicitly test accuracy gains from multi-contrast integration.

We also measure robustness to images of variable quality by synthetically corrupting a subset of images in a group. We evaluate using 500 synthetic brains with arbitrary contrasts, measuring average Dice over 35 anatomical labels. We find that all three mechanisms achieve similar overall accuracy on uncorrupted inputs (see Appendix Table 3). However, as shown in Figure 4C, attention interaction is markedly more robust to image quality: under corrupted inputs, Dice degrades by only $0.6 \pm 3.4\%$, compared to $4.6 \pm 4.5\%$ and $5.4 \pm 4.0\%$ for max- and mean-reduction, respectively.

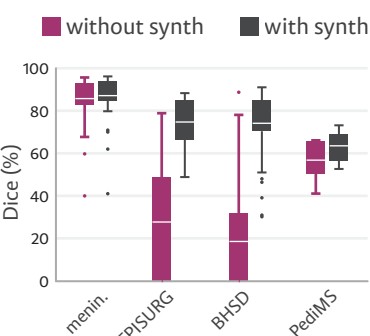

**Lesion Synthesis.** To assess the impact of synthetic lesion augmentation, we train a VoxelPrompt model without lesion synthesis and evaluate segmentation accuracy on the four held-out pathology datasets, following the same evaluation done in Section 4.1. As shown in Figure 5, lesion synthesis results in a mean 27.7% Dice increase across all four datasets, with the most substantial improvements for EPISURG and BHSD segmentation.

Figure 5: Training on synthetic lesions enables better zero-shot generalization on pathology datasets.

## 5 DISCUSSION

**Limitations and Future Work.** The VoxelPrompt vision and language networks are trained from scratch on template-generated prompts, which limits generalization to entirely novel prompts and tasks. As a result, VoxelPrompt as a vision system tackles common quantitative ROI-based neuroimaging workflows rather than open-domain instruction handling and general-purpose software engineering. However, this limitation can be addressed by future work training on more heterogeneous image-prompt datasets and integrating pretrained foundation language models that either generate more variable synthetic prompts or act as planners that use the trained VoxelPrompt model. Additionally, VoxelPrompt was trained on a combination of public brain imaging datasets and images we aggregated and annotated, but some task-specific datasets remain small. The training set can be extended further by inferring tasks from clinical images and their associated reports, which might better cover the complex edge-case pathologies seen in practice. We believe that our training strategy is generic and could be productively applied to radiology domains beyond neuroimaging.

**Conclusions.** We introduced VoxelPrompt, a vision-language system that can accurately address heterogeneous and complex radiological aims not possible with existing methods, as well as tasks that today require a multitude of specialized models and extensive manual user work. We demonstrated VoxelPrompt's capability to handle a wide range of complex, multi-image workflows, quantitative ROI analysis, and pathology-focused question-answering within a code execution system, which provides transparent end-to-end execution steps for clinically meaningful outputs. We anticipate VoxelPrompt's use in projects, *ad hoc* studies, and clinical pipelines, empowering biomedical users to adopt AI into their imaging workflows.

ETHICS STATEMENT

We have read and adhere to all terms of the ICLR Code of Ethics.

REPRODUCIBILITY STATEMENT

All details regarding model implementation and training data design are provided in Section 3 and Appendices A and B. Datasets used in this paper are publicly available and freely accessible, and preprocessing details are provided when used. References to all images gathered from *Radiopaedia* are provided in the supplementary material. We have attached our library of functions used by the VoxelPrompt agent as supplementary material and will make our weights and remaining code publicly available.

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

## A  MODEL IMPLEMENTATION DETAILS

We implement VoxelPrompt with PyTorch (Paszke et al., 2019) and use Python as the programming language of the code $k$. To support the wide range of imaging operations required by $\mathcal{T}$, we develop and use a PyTorch library of volumetric medical image utilities, called *Voxel*, attached in the supplementary zip file. A table summarizing core variables defined in the Methods section is provided below.

| variable | description |
|---|---|
| $\mathcal{V}$ | set of $n$ input volumes |
| $\mathcal{W}$ | set of $n$ output segmentation volumes |
| $m$ | multi-scale convolutional vision model |
| $m_{\text{enc}}$ | volume encoder component of $m$ |
| $\alpha$ | language model |
| $k$ | code (text) predicted by $\alpha(\cdot)$ |
| $b$ | token embedding dimension of $\alpha$ |
| $u$ | embedding token sequence input to $\alpha(\cdot)$ |
| $z$ | embedding token sequence representing outputs generated during code executing |
| $\phi$ | vision conditioning embedding sequence in $\mathbb{R}^{n \times b}$ |
| $\varepsilon$ | encoded volume embedding sequence in $\mathbb{R}^{n \times b}$ |

### A.1  LANGUAGE MODEL

We implement the language model $\alpha$ as a decoder-only transformer stack, using a randomly initialized LLaMA architecture (Touvron et al., 2023; Wolf et al., 2019) with 16 transformer layers, a hidden representation of size $b = 512$, a linear representation of size 1024, and 32 attention heads. We convert text into an embedding space by splitting character groups into tokens (from a vocabulary of size $\gamma$) and mapping them to a sequence of $\mathbb{R}^b$ features via an embedding matrix in $\mathbb{R}^{\gamma \times b}$. We use the pre-computed tokenizer released with LLaMA 2, with $\gamma = 32,000$.

The language model auto-regressively generates instruction embeddings $\varphi = \varphi^k \,\|\, \varphi^\phi$ based on the input $\mu$. We pass $\varphi^k$ through a fully-connected layer to obtain text token probabilities $P(k)$, and decode into code $k$ by choosing the maximum probability token at each sequence position. We pass $\varphi^\phi$ through a fully-connected layer to compute the vision network modulators $\phi$.

To split $\varphi^\phi$ and $\varphi^k$, we first transform $\varphi$ embeddings into a sequence of max-probability tokens. We extract $\varphi^\phi$ from all sequence positions that immediately follow special token `<MOD>`, and we extract $\varphi^k$ from all remaining positions. The agent $\alpha$ predicts `<MOD>` and subsequent $\varphi^\phi$ features after each volume encoding and generation function argument. We project $\varphi^\phi$ embeddings to $\phi$ using a fully-connected layer with 32 output channels and SiLU activation.

### A.2  PERSISTENT ENVIRONMENT

The language model predicts code to solve tasks using built-in Python functions, volumetric and morphological imaging functions from our *Voxel* package, as well as `encode` and `segment` functions that execute the vision model. In the Python execution environment, we predefine a variable corresponding to each volume $v$. As the code $k_i$ is executed, new variables are defined and retained in the environment, which persist across steps. Text outputs generated during the execution process (e.g., variable print statements) are embedding into a representation $z_i$ as feedback in the next state $\mu_{i+1} = \mu_i \,\|\, \varphi_i \,\|\, z_i$. When an `encode` function is executed on $\mathcal{V}$, we concatenate the resulting $\varepsilon$ into the feedback embeddings $z$.

### A.3  VISION SUBNETWORK

We implement $m$ as a six-level UNet-like model (Ronneberger et al., 2015). Each level consists of a 3D convolutional layer, optionally followed by a conditioning, cross-volume interaction layer, as defined in Section 3.1. The interaction transformer has a 32-dimensional hidden size and uses a residual connection. All layers are activated with SiLU. The spatial outputs at each level are

channel-normalized with a group size of four, then max-pooled by a factor of two. Convolution kernels have size $3^3$, with 32 output channels at the top resolution level and 96 output channels at all other levels. Lastly, we apply the sigmoid activation to generate binary segmentation maps. If multiple volumes from a single scan session are passed as input to VoxelPrompt, we compute a merged, session-specific segmentation by extracting the max values across outputs corresponding to each session.

To reduce the spatial feature dimensions into the final vector encodings $\varepsilon$, for each input volume passed through $m_{enc}$, we extract the corresponding output activation map at the lowest resolution layer. We reduce spatial feature activations across all dimensions using a global max-pooling operator, pass pooled features through a fully-connected projection layer, and stack the $n$ embedding representations to obtain $\varepsilon \in \mathbb{R}^{n \times b}$.

### A.4 Optimization

We train VoxelPrompt using the Adam optimizer (Kingma & Ba, 2014) with an initial learning rate of $10^{-4}$, a batch size of one, and 10 gradient accumulation steps on an NVIDIA A100 GPU. We halve the learning rate after $10^5$ steps with no improvement in validation accuracy, stopping training after four sets of learning rate updates. We set the volume loss weight $\lambda = 0.1$.

## B Training Data Details

### B.1 Task Categories

Quantitative and morphometric processing tasks often involve image segmentation, followed by downstream steps to compute ROI measures. For example, some tasks involve removing, extracting, or cropping the field of view (FOV) around a segmented region. Others use segmentations to compute ROI-specific statistics of image signal intensities (e.g., mean intensity). Morphological tasks can involve analyzing ROI shape and compute total volume, bounding box dimensions, or the maximum height, width, and depth of a segmented structure.

We also include tasks that compute and compare such metrics across multiple segmentations. For example, longitudinal tasks measure change in ROI properties across a series of scan sessions, and multi-region tasks compare metrics from different ROIs in a single scan session. For tasks involving analyzing longitudinal changes, input volumes span two imaging sessions.

We also include question-answering tasks pertaining to pathology property analysis. These include: (1) identification of the anatomical location of a lesion, (2) characterization of an ROI's signal intensity as hyperintense, hypointense, or isointense relative to surrounding tissue, (3) detecting restricted diffusion from paired DWI and ADC maps, or (4) assessing post-contrast lesion enhancement. We enumerate the high level task categories generated for training and evaluation in Table 1. For tasks measuring ROI extent, sampled subtasks can include linear measurements along the sagittal, coronal, or axial planes or full 3D bounding box dimension measurements.

### B.2 Image Preprocessing and Augmentation

For each image volume, we normalize intensities within the range $[0, 1]$, conform the data layout to a right-anterior-superior (RAS) orientation, and crop the field of view to a 20 mm margin around the cranial cavity. We co-register all images acquired from each subject using Hoffmann et al. (2024).

In training, we randomly sample up to 8 (or max available) images corresponding to a scan session. We augment images by applying random affine transformations, spatial intensity distortions (bias field simulations, spatial smoothing, $k$-space corruptions), exponential scaling, lateral anatomical flipping, cropping, anatomical masking, and voxel resizing. We take advantage of our resolution-agnostic vision network and randomly sparsify training data to reduce voxel throughput and significantly reduce total train time. This random sparsification is performed by sampling slice separations from the range $[1, 6]$ mm or by cropping the field of view. We ensure that the target ROI, if applicable, is not removed during this process. Volume sparsification is performed with 50% probability for each sample or when total input voxels exceeds a preset threshold to prevent device memory errors.

Table 1: High-level task categories generated for training and evaluation.

| category | example prompt | output type |
|---|---|---|
| segmentation | *delineate the ⟨roi⟩* | segmentation |
| volume analysis | *determine the size of the ⟨roi⟩* | volume |
| volume change | *how much has the ⟨roi⟩ progressed?* | volume diff. |
| volume comparison | *compute the bilateral assymetry of ⟨roi⟩* | volume diff. |
| relative volume analysis | *what percent of ⟨roi⟩ is hyperintense?* | volume diff. |
| extent analysis | *what is the lateral width of the ⟨roi⟩?* | distance |
| extent change | *how much has the ⟨roi⟩ widened?* | distance diff. |
| center of mass | *locate the ⟨roi⟩ center of mass* | point coordinate |
| intensity analysis | *compute the mean ⟨roi⟩ density* | mean intensity |
| intensity comparison | *how hyperintense is ⟨roi⟩ relative to ⟨roi⟩?* | mean intensity diff. |
| intensity change | *determine progressive signal loss of ⟨roi⟩* | mean intensity diff. |
| location | *where is the ⟨roi⟩?* | natural language |
| enhancement | *does ⟨roi⟩ enhance post contrast?* | natural language |
| diffusion restriction | *does ⟨roi⟩ restrict diffusion?* | natural language |
| signal intensity | *characterize signal properties of ⟨roi⟩* | natural language |
| image cropping | *crop the image around ⟨roi⟩* | volume(s) |
| tissue removal | *remove ⟨roi⟩ signal from the image* | volume(s) |

## B.3 ANATOMICAL DATASET DETAILS

In addition to the pathology datasets outlined in 3.2, we generate segmentations for whole-brain anatomical structures on images from the FSM (Greve et al., 2021), OASIS (Marcus et al., 2007; LaMontagne et al., 2019), Mind Brain Body (Babayan et al., 2019), IBC (Pinho et al., 2018), CER-MEP (Mérida et al., 2021), and Forrest Gump (Hanke et al., 2014) cohorts. We select high-quality acquisitions and thoroughly inspect and correct errors in the label maps. Additionally, we make use of multiple image atlases with precomputed segmentations (Adil et al., 2021; Pauli et al., 2018).

## B.4 ANATOMICAL STRUCTURES

We use segmentations of various anatomical classes, listed below. Bilateral brain structures are defined by two distinct hemisphere-specific labels.

- Global tissue classes include the brain, dura, skull cavity, cerebrum, cerebral white matter, cerebral cortex, brainstem, cerebellum, ventricular system, and cerebral spinal fluid (CSF).

- Brain sub-structure labels include the amygdala, nucleus accumbens, hippocampus, thalamus, caudate, putamen, dorsal striatum, globus pallidus (externus and internus), basal ganglia, hypothalamus, fornix (body, crus, and column), mammillary body, septal nucleus, subthalamic nucleus, habenula, ventral pallidum, extended amygdala, red nucleus, anterior and posterior commissures, pars compacta, pars reticulata, parabrachial pigmented nucleus, ventral tegmental area, fimbria, septum pellucidum, tectum, pineal gland, superior and inferior colliculus, cerebral peduncle, medullary pyramid, medial lemniscus, cerebellar peduncle (superior, middle, inferior), cerebellar gray matter, and cerebellar white matter.

- Ventricular sub-structure labels include the lateral ventricle, inferior lateral ventricle, posterior lateral ventricle, anterior lateral ventricle, atrium, third ventricle, fourth ventricle, interventricular foramen, and cerebral aqueduct.

- Cortical sub-region labels include the frontal lobe, parietal lobe, temporal lobe, occipital lobe, cingulate cortex, insular cortex, anterior cingulate cortex, caudal anterior cingulate cortex, rostral anterior cingulate cortex, posterior cingulate cortex, isthmus cingulate cortex, frontal pole, middle frontal gyrus, caudal middle frontal gyrus, rostral middle frontal gyrus, superior frontal gyrus, inferior frontal gyrus, pars opercularis, pars orbitalis, pars triangularis, lateral orbitofrontal cortex, medial orbitofrontal cortex, precentral gyrus, paracentral lobule, inferior parietal lobule, superior parietal lobule, supramarginal gyrus, precuneus, postcentral gyrus, entorhinal cortex, fusiform gyrus, parahippocampal gyrus,

temporal pole, inferior temporal gyrus, middle temporal gyrus, superior temporal gyrus, transverse temporal gyrus, cuneus, lingual gyrus, and pericalcarine cortex.

## B.5 RADIOPAEDIA DATA

We download and annotate 101 patient cases from Radiopaedia, a radiology reference at *https://radiopaedia.org*. Each case includes text-based notes and scans in the form of 2D image slices. We reconstruct volumetric data by stacking these slices and estimating an affine matrix to map voxel coordinates in world space. We compute this mapping by registering the image to an average template. The cases used are provided in the supplementary `case-references.md` file.

## B.6 LESION SYNTHESIS

To extend the range of pathological features observed during training, we synthesize brain lesions with variable characteristics using a model-based domain randomization technique (Gopinath et al., 2024). Images spanning diverse acquisition types from 26 subjects in the OASIS, Mind Brain Body, and CERMEP datasets are paired with whole-brain anatomical segmentation maps as the basis for this process.

For each synthetic lesion, we first sample parameters describing anatomical location, dimensions, intensity relative to surrounding tissue, and texture. As illustrated in Figure 7, volumetric multi-scale Brownian noise is generated with a signal fall-off matched to the sampled lesion dimensions, then thresholded to produce lobulated structures. These shapes define lesion boundaries, which are further constrained by anatomical maps. For example, parenchymal lesions are restricted to white and gray matter, while ventricular lesions are restricted to cerebrospinal fluid spaces. Lesion interiors are inpainted into the native image using randomly generated textures derived from Perlin noise. The mean intensity of the inpainted lesion is determined by the underlying healthy image signal and sampled relative intensity shift.

**Example Prompt Template**

{compute} {volumetric growth} {of the} {glioma, hyperintense, left insular lobe}

| what is | the change in mass | of the | abnormality |
| estimate | any size difference | of the | hyperintensity |
| derive | progressive growth | of the | mass near insular cortex |
| quantify | the volume delta | of the | mass occupying lesion in the left hemisphere |
| compute | size increase | of the | hyperintense, insular neoplasm |
| evaluate | any growth in volume | of the | glioma (hyperintense) in the left insular lobe |

ROI description detail

**Examples of Generated Prompts Across Tasks**

delineate the amygdalar nuclei on the left side
characterize the density of the brainstem lesion on the right
has the right parietal lesion signal decreased?
what is the size of the right pallidum?
estimate left caudate volume
does the frontal lobe mass restrict diffusion?
can you show me the dorsal striate nucleus?
what is the change in extent of the MCA left ischemic infarct?
segment the white matter hyperintensities
estimate the anatomical location of the hyperintense mass
compute relative hypointensity of the chronic infarction to white matter
crop the image to the right-side telencephalon
what are signal characteristics of the lesion within the cerebellopontine angle?
strip signal that is not left cerebellar cortex
isolate only the lateral ventricular mass cystic component

evaluate total tumor volume change over time
crop 5 mm around the thalamic nuclei (left)
compute a left inferior parietal lobule segmentation
what is the length of fourth ventricle in the coronal direction?
measure the volume of the lesion hyperintense foci
compute the dimensions of the left dorsal striatum
derive volume difference between left and right fusiform gyrus
characterize the signal enhancement of the glioma post contrast
quantify growth of the hippocampus region on the left
estimate the superior-inferior length of the neurocytoma
derive the volume of the cavum veli interpositi
where is the hypodense tumor?
what is the vascular territory of the lesion?
calculate the volume of the solid tissue component of the lesion
highlight the putamen within the right hemisphere

Figure 6: **Top:** An example template used to generate random prompts with varied combinations of words and phrases and varied degrees of ROI description detail. **Bottom:** Examples of generated training prompts across numerous task categories.

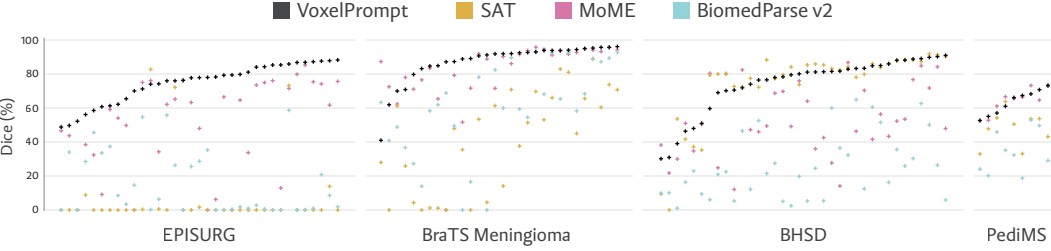

generate noise → attenuate signal → binary threshold → obtain lesion segmentation mask → in-paint lesion on scan

Figure 7: Schematic of the lesion synthesis procedure. A lesion shape is first generated by attenuating and thresholding Brownian noise. The resulting segmentation map is resampled into the target image space, with size and position determined based on anatomical priors. The lesion is in-painted by pasting the tissue mask into the image with procedurally-generated texture and mean signal intensity based on randomly selected relative tissue characteristics.

We also synthesize multiple lesions per subject with varying properties, providing negative examples for VoxelPrompt to learn to differentiate abnormalities based on descriptive features. In addition, we simulate heterogeneous lesions by superimposing secondary Brownian noise–derived masks within an existing lesion, producing intra-lesional components or heterogeneous signal profiles.

## C EXPERIMENTAL DETAILS AND DATA

### C.1 SEGMENTATION EVALUATION DATA

The pathology test set used for specialist model ablations in Section 4.2 includes 206 images of infarcts (206 subjects), 1,376 images of gliomas (344 subjects), 1,386 images of edema (347 subjects), 24 images of cysts (11 subjects), 16 images of papilloma (6 subjects), 21 images of meningioma (8 subjects), and 40 images of white matter hyperintensities (20 subjects). The anatomical evaluation set consists of 108 images (40 subjects), from the dataset defined in Appendix B.3, and is used for both the whole-brain segmentation experiments in Section 4.1 and the single-task ablation analysis in Section 4.2. End-to-end evaluation of bilateral ventricle asymmetry uses 76 subjects from ADNI.

### C.2 ZERO-SHOT LESION SEGMENTATION BASELINES

For the zero-shot segmentation experiments, we evaluate baseline performance across diverse input configurations to ensure fairness and avoid bias. For both BiomedParse v2 (Zhao et al., 2025a) and SAT (Zhao et al., 2025b), we explore a range of prompting strategies on each dataset, following the formats recommended in the original repositories or used during their training. We test multiple levels of pathology classification terminology and adopt the phrasing that yields the best performance. BiomedParse v2 does not specify a preferred anatomical orientation, so we evaluate across all possible orientations and report the best-performing one. We also find that BiomedParse v2 does not benefit from resampling inputs to isotropic resolution. The MoME (Zhang et al., 2024) baseline

Figure 8: Per-subject accuracy in terms of Dice score performance for zero-shot lesion segmentation. Subject indices are sorted along the x-axis in ascending order of VoxelPrompt prediction accuracy. We find that VoxelPrompt consistently achieves high-quality segmentation across all evaluation datasets and lesion types considered.

requires skull-stripping and affine alignment to the MNI template prior to prediction. We follow this preprocessing and report MoME's results in the original coordinate system.

For BioMedParse v2, we explored a range of prompt formats based on the model release documentation and examples used during training. We varied syntactic structure, testing prefixing strategies (e.g., "imaging of ⟨roi⟩", "MRI imaging of ⟨roi⟩", "CT imaging of ⟨roi⟩", "scan of ⟨roi⟩", "image of ⟨roi⟩") and testing prompts that used only the ROI terminology without additional context. For SAT, we evaluated the prompting templates provided in the original training documentation as well as variants containing additional clinical or imaging descriptors. For both models, we tested multiple ROI names, including plural forms and variants prefixed with "brain", as well as terminology not necessarily matched to the ground-truth pathology category. These included: abnormality, pathology, disease, lesion, tumor, mass, meningioma, glioma, glioblastoma, hyperintensity, hypointensity, hyperdensity, hypodensity, leukoaraiosis, white matter hyperintensity, edema, cavity, resection, cyst, necrosis, hemorrhage, intracranial hemorrhage, bleed, blood, infarct, and stroke (with acute, subacute, or chronic modifiers where applicable). We selected the prompts that maximized evaluation Dice performance for these baselines.

The optimal prompts used as input for each language-conditioned method are below:

|  | VoxelPrompt | BiomedParse V2 | SAT |
|---|---|---|---|
| BraTS menin. | segment the hyperintense mass | MRI imaging of a brain tumor | meningioma |
| EPISURG | segment the hypointense lesion | MRI imaging of a brain lesion | stroke |
| BHSD | segment the hyperdense lesions | CT imaging of brain lesions | intracranial hemorrhage |
| PediMS | segment hyperintensities | MRI imaging of lesions | white matter hyperintensities |

In Figure 8, we show per-subject lesion segmentation performance for all methods. VoxelPrompt consistently outperforms the baseline foundation models for brain pathology segmentation.

### C.3  END-TO-END ANALYSIS PROMPTING

The following prompts were used as inputs to VoxelPrompt for the end-to-end performance analysis in Section 4.1: "*compute volume of the ⟨roi⟩*" for volume estimation, "*compute volume asymmetry of the lateral ventricle*" for ventricular asymmetry estimation, "*compute lateral width of the ⟨roi⟩*" for pathology width estimation, and "*compute mean density of the hyperdense lesions*" for hemorrhage density estimation.

### C.4  SCAN GEOMETRY SAMPLING

To evaluate the efficiency improvements provided by the native-resolution vision network, we sample test images with spatial geometries drawn from distributions representative of clinical MR and CT brain acquisitions. Image dimensions are sampled uniformly around a $155 \times 190 \times 165$ mm field of view, with a max deviation of $\pm 15$ mm in each dimension. This range is derived from the 10th–90th percentile values of image sizes in our preprocessed datasets.

To reflect real-world voxel resolutions, we consider imaging modalities most frequently collected in a standard clinical brain imaging session. For each acquisition type, we define uniform sampling ranges for in-plane resolution and slice separation, based on standard protocol guidelines and empirical resolution distributions observed in our dataset. Voxel spacings are clamped to a minimum of 0.8 mm. During each experimental sample, we randomly select a field of view, acquisition type, and resolution from these distributions, and randomly populate the volumes with Gaussian noise. This distribution, outlined in Table 2, is not exhaustive, but is designed to provide a representative coverage of common acquisitions sufficient to evaluate the benefits of native-resolution processing.

### C.5  IMAGE SYNTHESIS FOR EVALUATING STREAM INTERACTION

Brain image synthesis techniques are used to train neuroimaging models that are robust to acquisition variability and anatomical differences (Gopinath et al., 2024). These approaches employ domain randomization methods (Tobin et al., 2017), in which whole-brain anatomical segmentations are

|  | in-plane spacing (mm) | slice separation (mm) |
|---|---|---|
| T1-weighted (isotropic) | 0.8 – 1.2 | iso. |
| T2-weighted | 0.8 – 1.0 | 3.0 – 5.0 |
| FLAIR | 0.8 – 1.0 | 3.0 – 5.0 |
| diffusion-weighted (DWI) | 1.5 – 2.5 | 2.0 – 3.5 |
| gradient-echo (GRE) | 0.8 – 1.2 | 4.0 – 6.0 |
| perfusion MRI | 1.5 – 2.5 | 4.0 – 6.0 |
| susceptibility-weighted (SWI) | 0.8 – 1.0 | 1.5 – 3.0 |
| CT (isotropic) | 0.8 – 1.0 | iso. |
| CT (thick slice) | 0.8 – 1.0 | 3.0 – 6.0 |

Table 2: Distribution of common scan geometries sampled for efficiency ablations.

mapped to randomized tissue intensities, warped by spatial transformations, and augmented with simulated artifacts. The resulting synthetic images extend beyond the realistic range of clinical scans, enabling models to generalize across diverse real-world data and tasks (Dey et al., 2024).

We adopt this strategy as a controlled framework for evaluating VoxelPrompt's ability to integrate complementary information across arbitrary numbers of input volumes. By generating multiple synthetic images from a single anatomical label map, we test whether segmentation accuracy improves as additional inputs are provided.

For stream-interaction evaluation, we employ a standard image synthesis protocol widely used in brain imaging (Gopinath et al., 2024). Briefly, for each evaluation sample, we sample a whole-brain anatomical segmentation map from a set of OASIS subjects. To generate an individual image from this segmentation, each label is assigned an intensity distribution defined by Gaussian parameters sampled uniformly, following a classical Bayesian segmentation formulation. Voxel labels are then recoded into grayscale values drawn from their respective label distributions, producing synthetic images. Finally, we apply random artifact simulations, including spatial blurring, additive noise, and bias-field distortion.

To generate corrupted images, we synthesize random label maps from multi-scale Brownian noise. Between 10 and 20 noise maps are generated, and voxels are assigned to the index of the maximum-valued map. This synthetic label map is then converted to an image using the same label-to-intensity procedure described above.

## C.6 STREAM INTERACTION VARIANTS

We compare our attention-based stream interaction module with two reduction-based variants commonly used in multi-input medical image analysis (Butoi et al., 2023). In these variants, features are aggregated across input streams by mean or max pooling along the stream dimension. The resulting global feature representation is then concatenated channel-wise with the original stream-specific features. This combined feature map is passed through a linear projection layer whose output dimensionality matches that of the block input, ensuring compatibility with the downstream network. We test both mean and max-reduction as alternative aggregation operations to our proposed attention mechanism.

To compare interaction mechanisms, we implement vision models matching the VoxelPrompt architecture but without language-conditioning blocks, varying only the stream interaction module. Models are trained on synthetic data (Appendix C.5) using multi-class Soft Dice loss to segment 35 anatomical brain structures. At each training step, between one and three images corresponding to a single segmentation target are sampled, with a 10% probability of including a corrupted image (described in Appendix C.5).

For evaluation, we generate 500 synthetic image sets, each containing three images derived from a held-out test set of 50 subjects and a predefined subset of corrupted images. We evaluate model performance by varying the number of images provided as input from each set and measuring Dice overlap between predicted and reference segmentations.

Table 3: On *uncorrupted* input image sets, attention- and reduction-based stream interaction methods result in similar model segmentation accuracy (Dice), which improves for all methods as image inputs are added for a single forward pass.

| Method | 1 image | 2 images | 3 images |
|---|---|---|---|
| attention | $83.7 \pm 2.8$ | $85.8 \pm 1.8$ | $86.9 \pm 1.4$ |
| max-reduction | $83.5 \pm 2.6$ | $85.8 \pm 1.8$ | $87.1 \pm 1.5$ |
| mean-reduction | $83.0 \pm 2.7$ | $85.7 \pm 1.8$ | $86.7 \pm 1.6$ |

## D  PATHOLOGY CHARACTERIZATION

We evaluate the ability of VoxelPrompt to produce a natural language characterization of image features. We focus on five pathology-based visual question-answering tasks (also used during training). These involve classifying lesions based on (1) signal intensity, (2) broad cerebral location, (3) stroke-affected vascular territory, (4) diffusion restriction, and (5) post-contrast enhancement.

**Data.** For each of these tasks, we curate a subset of held-out subjects with relevant features, while ensuring equal representation of possible classification categories in each subset. In total, the evaluation set consists of 102 cases, with 26 images for characterizing lesion signal intensity (26 subjects), 112 images for identifying lesion cerebral location (30 subjects), 16 images for identifying infarct vascular territory (12 subjects), 28 images for detecting diffusion restriction (14 subjects), and 40 images for detecting post-contrast enhancement (20 subjects).

**Evaluation.** During evaluation, we consider a prediction as correct if the output natural language response exactly matches the expected characterization. Using a paired *t*-test, we compare the VoxelPrompt per-task classification accuracy to that of multiple baselines.

**Baselines.** We compare VoxelPrompt to a set of classifier benchmarks, each trained for one of the five pathology characterization tasks in $\mathcal{T}$. As opposed to using language, the single-task benchmark models directly predict label probabilities for a fixed set of task-specific characterizations. We implement these models using the architecture of $m_{\text{enc}}$, with $\phi$ mixing layers replaced. We reduce the spatial dimensions of the deepest encoder layer output using a global max operator, then compute the maximum value over all input volume streams. To compute classification probabilities for $n$ possible descriptions, we pass the stream-pooled features to a fully-connected layer with $n$ output channels and softmax activation. During benchmark optimization, we use the categorical cross-entropy loss on these predicted probabilities.

We also compare VoxelPrompt to RadFM (Wu et al., 2025), a publicly released, state-of-the-art architecture for 3D medical visual question answering. In our preliminary experiments, we find that the pretrained RadFM *cannot generalize to any* of the neuroimaging tasks used in this experiment. Therefore, for fairness, we *fine-tune* RadFM on our subset of pathology characterization tasks, using the training code released with their pretrained model weights. To fit the optimization within 80 GB of GPU memory, we keep only the first eight hidden transformer layers of the language model and do not modify any other model components. As required by the vision transformer, we resize all

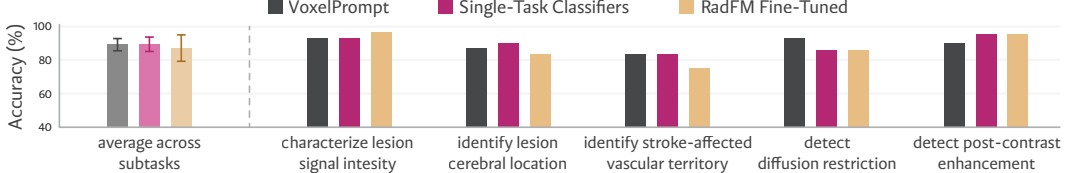

Figure 9: Accuracy of pathology characterization using natural language for five separate classification subtasks. Average subtask accuracy is shown on the left. VoxelPrompt (black) parallels the performance of individually-trained, single-task classifiers (purple) and a fine-tuned RadFM model (blue) – a state-of-the-art method for 3D visual question-answering.

input volume spatial dimensions to the nearest multiple of $32 \times 32 \times 4$. During fine-tuning, we use only the expected language response (without code) as the target text.

**Results.** Figure 9 shows that VoxelPrompt achieves a mean classification accuracy of $89.0 \pm 3.6\%$ over all tasks, matching the performance of the single-task benchmarks ($89.3 \pm 4.2\%$) as well as the fine-tuned RadFM model ($87.1 \pm 7.9\%$). These results demonstrate that VoxelPrompt can achieve language-based image characterization with comparable performance to specialized classification and medical vision-language architectures, while also able to perform the wide variety of tasks described in the main experiments.

