# OpenReview forum: "VoxelPrompt: A Vision Agent for End-to-End Medical Image Analysis"
_ICLR.cc/2026/Conference — Submitted to ICLR 2026_

### Official Review · Reviewer_DgCn · 2025-10-19

**Soundness:** 2
**Presentation:** 2
**Contribution:** 3
**Rating:** 2
**Confidence:** 3

**Summary:**

This work focuses on improving generalization of language-vision medical image analysis to diverse practical clinical use cases across real-world lesions types. Specifically, VoxelPrompt is an agent-based approach that coordinates outputs from vision and language models to produce executable code that performs tasks such as ROI measurements, image manipulation, language characterization, and analyses across multiple ROIs, multiple acquisitions, and multiple visits. The authors also propose a CNN model that integrates information from language prompts and processes images in native resolution. Furthermore, they propose a procedure for constructing a dataset that that improves robustness to lesion types across different datasets.

**Strengths:**

- The use of agents to produce code seems like a promising approach to generating interpretable operations on medical images.
- The evaluations seem to consider a broad range of relevant use cases
- The lesion synthesis procedure seems to be a novel approach.
- This work also highlights an important need for free-form workflow benchmarks that capture common practitioner use-cases

**Weaknesses:**

- **Unclear notation**:
    - Section 3.1: The notation that is introduced is not precise. What exactly is $V, p, a, \Omega, W$, etc.. Are these vectors, scalars, matrices, functions, etc.?
        - Line 186: $\phi$ is defined as “image-specific latent instruction embedding”. However, there is no further detail about what this object is or where it comes from. Similarly, where does $\phi_s$ (line ~209) come from?
        - Line 210: The shape notation $\mathbb{R} ^{S,c}$ is not standard. I think you mean $\mathbb{R} ^{S \times c}$, but it is not clear in the text.
- **Unclear description of model architecture**:
    - Line 207: I’m not sure I understand the role of “streams”. How are these different than the standard intermediate activations from a transformer layer, where inputs are processed separately before fusion with an attention mechanism?
    - Line 215-222: The section on native space processing is unclear. What is the base architecture and what is the upsampling and downsampling arm referring to? It would also be helpful to elaborate on what “common geometry” (line 222) mean in this context. What exactly is being updated to adapt the model for different resolutions? What mechanism allows enables the sharing of model weights across these different resolutions?
- **Unclear description of Dataset generation procedure**:
    - Overall, section 3.2 is quite vague. In general, it is not clear what the exact tasks are, what their inputs and outputs are, how they are generated, or what are the parameters that can be varied during generation. Concretely, here are a few examples:
        - Line 244-250: In section “training code for quantitative ROI processing”, it is unclear if these tasks are considered separately or if they are combined together. How are multiple tasks sampled during training?
        - Line 252: what exactly are the relevant metrics? Please be specific. What is the objective function that you are optimizing?
        - Line 260: In section “training code for question answering”, Its not clear how correct natural language text response is generated.
        - Line 263: how did you construct these templates? Can you provide some examples of what these templates look like?
        - Line 266: How do you make sure that the generated prompt doesn’t produce invalid combinations?
- **Unclear description/motivation for the evaluation**:
    - Line 311: It is unclear what zero-shot lesion segmentation is intended to evaluate. Could you clarify what type of generalization the held-out pathology datasets are designed to test? Are we testing for generalization to unseen diseases or same diseases but unseen populations or unseen tasks, or something else?
    - Section 4.2. What is specialist network referring to? Can you provide a citation here?
    - Could the authors clarify how the evaluations in Section 4 validate the use cases presented in Figure 1? At present, it is not clear whether VoxelPrompt successfully accomplishes its intended use cases.
- **Insufficient baselines**. For certain evaluations, the authors evaluate compare against only a single baseline which is insufficient in evaluating how the proposed approach compares to the existing literature.
    - Fig 3D: Why not compare performance against other VLMs? My intuition says that other VLMs will achieve a similar runtime improvement when compared against FreeSurfer.
    - Fig 3E: Same comment here, why did you only choose to benchmark against SynthSeg? Why not compare against other segmentation or VLM models?

**Questions:**

Please see weakness section for the majority of questions. Here are a few additional questions:
1. In line 198, How do you guarantee that the feedback procedure does not result in an infinite loop?
2. How well does this approach generalize to new tasks, especially more complex multi-step tasks? It seems like the training procedure is quite specific to simple tasks that are provided during training. Is the dataset generation approach scalable?
3. What is the effect of synthetic lesion training? Can you do an ablation to show that the proposed approach is effective?

---

> ### Author Response · Authors · 2025-11-25
> **Response to Reviewer DgCn (part 1 of 4)**
>
> We deeply appreciate the reviewer's attention to detail regarding our technical descriptions and experiments. Their feedback has been integrated broadly across the paper.
>
> ## Notation
>
> Thank you for finding these oversights. To clarify variable definitions we have revised the Methods text. Further, we now include an Appendix Table 1 that maps each core variable to its description.
>
> > *Section 3.1: The notation that is introduced is not precise. What exactly is $V, p, \alpha, \Omega, W$, etc.. Are these vectors, scalars, matrices, functions, etc.?*
>
> The sets $V, W$ contain input image volumes and output segmentation volumes, respectively. A volume consists of a feature tensor in $\mathbb{R}^{c \times w \times h \times d}$ (with $c$ channels and $w,h,d$ spatial dimensions) and a world-coordinate matrix in $\mathbb{R}^{4 \times 4}$ specifying voxel spacing and position. We denote $\Omega$ as the Python environment in which code predicted by VoxelPrompt is executed (we have removed this variable from the text for simplicity). The notation $\alpha(\cdot)$ defines the language-model forward pass, and $p$ represents the input text prompt string (also removed for simplicity). We now update the description of these and the remaining variables in the revised text.
>
> > *Line 186: $\phi$ is defined as “image-specific latent instruction embedding”. However, there is no further detail about what this object is or where it comes from. Similarly, where does $\phi_s$ (line ~209) come from?*
>
> We should have described this better. The matrix $\phi \in \mathbb{R}^{n \times b}$ contains $n$ token embeddings of size $b$ produced by the language model and used to condition the vision model, for $n$ input volumes. We now clarify this in the Methods section and simplify the variable definition by removing the subscripted $\phi_s$.
>
> > *Line 210: The shape notation $\mathbb{R}^{S,c}$ is not standard*
>
> Thank you for catching these typos. We have standardized them to $\mathbb{R}^{a \times b}$ format.
>
> ## Model architecture description
>
> > *Line 207: I’m not sure I understand the role of “streams”. How are these different than the standard intermediate activations from a transformer layer, where inputs are processed separately before fusion with an attention mechanism?*
>
> We have now taken out the *streams* terminology, which we agree was confusing. We clarify that our layers are similar to transformer layers, in that multiple inputs are first processed independently and then fused through an attention mechanism. However, unlike transformers, we do not use positional encodings since a group of input volumes (e.g., multiple scan modalities in a session) need not have a sequence order.
>
> In VoxelPrompt, each input volume (e.g., a different acquisition or timepoint) in an input set is processed in parallel through the same convolutional network, with weights shared across the set. After each convolutional block, we interact all features of the set via cross-attention applied across volumes at matched spatial indices, enabling features at corresponding spatial locations to exchange information across inputs. This is conceptually similar to general transformer layers without positional encoding, which operate on sets of *tokens*, and apply a *full* self-attention involving the entire tokens.
>
> We have now updated the technical description of this mechanism in the paper.
>
> > *Line 215-222: The section on native space processing is unclear. What is the base architecture and what is the upsampling and downsampling arm referring to?*
>
> We agree that this explanation was not clear enough, and have clarified and expanded it in Section 3.1 to better detail how information from multiple inputs is processed and integrated into a shared spatial representation.
>
> Briefly, our base vision architecture is a convolutional U-Net. The downsampling and upsampling arms refer to the encoder and the skip-connected decoder components, respectively. We insert language-conditioning, as well the voxel-wise cross-attention layers between inputs, after convolutional blocks.

---

> > ### Author Response · Authors · 2025-11-25
> > **Response to Reviewer DgCn (part 2 of 4)**
> >
> > > *It would also be helpful to elaborate on what “common geometry” (line 222) mean in this context. What exactly is being updated to adapt the model for different resolutions? What mechanism enables the sharing of model weights across these different resolutions?*
> >
> > Thanks for raising this. Handling this issue is an important part of the method and it is clarified in the revision. First, each scan in an input set (e.g. multiple scan modalities) has its own geometry (shape, resolution, and positioning). "Common geometry" is a neuroimaging term describing a shared spatial grid, to which multiple scans are aligned so that corresponding anatomical locations match voxel-to-voxel. Existing neuroimage analysis methods usually align each input to the higher resolution common geometry before processing all inputs, but this often requires substantial memory and processing.
> >
> > In VoxelPrompt, each input is processed through the same convolutional network in parallel, in its own native geometry, which often requires a much smaller footprint (Figure 4B). To execute feature interaction between images at each layer, we move the intermediate activation of each image in the input set to the common geometry space with isotropic voxel spacing. We then perform cross-attention across volumes at matched spatial indices, and interpolate the result back to each individual feature space.
> >
> > ## Dataset generation procedure
> >
> > > *Overall, section 3.2 is quite vague. In general, it is not clear what the exact tasks are, what their inputs and outputs are, how they are generated, or what the parameters that can be varied during generation.*
> >
> > To make this more concrete, we now substantially expand Section 3.2 to describe the task construction process and we include a list of all task categories in a new Appendix B.1. The updated section specifies how each task involves a manually constructed framework that samples relevant images, ROIs, prompts, and annotations, and instantiates a task "solution" in training with deterministic target output code and natural language.
> >
> > > *Line 244-250: In section “training code for quantitative ROI processing”, it is unclear if these tasks are considered separately or if they are combined together. How are multiple tasks sampled during training?*
> >
> > For each training iteration, we construct a single training task by sampling one task category from our task space $\mathcal{T}$, such as volumetric comparison or intensity change analysis. We now explicitly define and list high level task categories in Appendix B.1. We then sample predefined task-specific components, such as relevant regions of interest, an input image and segmentation pair (if appropriate for that task), and an input prompt, and we automatically generate the corresponding ground-truth text output. We sample a different task at each iteration in training. We have substantially expanded Section 3.2 to address these previously unclear details.
> >
> > > *Line 252: what exactly are the relevant metrics? Please be specific. What is the objective function that you are optimizing?*
> >
> > Metrics here mean "measures" provided as output to the user, like the volume, the difference in volumes, or the width of a mass. To be concrete, we added the list of expected measures for each task category to Appendix B.1.
> >
> > To clarify, although these measures are generated by the system, they are not used in the training objective. As described in Section 3.1, the loss function penalizes differences between predicted and expected output code, as well as between predicted and expected segmentation maps. Downstream measures are then generated from predicted segmentations.
> >
> > > *Line 260: In section “training code for question answering”, Its not clear how correct natural language text response is generated.*
> >
> > We expand the description of this process in Section 3.2. During training, we generate the ground truth natural language response for characterization-based tasks using a task-specific template, which converts predefined pathology annotations into a structured description. Concretely, the template contains semantic slots (e.g., {location}, {side}) that are filled by directly mapping each annotation field to its corresponding phrase set. These templates generate deterministic outputs. For example, for a tumor that is annotated as "parietal" and "left" in our data, the target output will be formatted as "left parietal lobe."

---

> > > ### Author Response · Authors · 2025-11-25
> > > **Response to Reviewer DgCn (part 3 of 4)**
> > >
> > > > *Line 263: how did you construct these templates? Can you provide some examples of what these templates look like? Line 266: How do you make sure that the generated prompt doesn’t produce invalid combinations?*
> > >
> > > We now provide an example template in Appendix Figure 6 and expand the description in Section 3.2.
> > >
> > > To construct templates from scratch, we first write an abstract sentence representation for each type of task, using grouped semantic slots that capture the semantic roles. For example, a core template may take the form "\{compute\} \{change\} \{of\} \{ROI\}" where each slot corresponds to a curated set of interchangeable phrases (e.g., {compute}: {compute, estimate, measure}; {change}: {change, growth, difference}). During training, prompts are generated by randomly substituting each slot with one element from its associated set.
> > >
> > > We construct a different template for each task. Although the final prompts appear diverse, the underlying templates are highly structured. Semantic slots are restricted to context-appropriate synonym groups, and only syntactic rearrangements that preserve meaning are allowed. This design minimizes semantically incompatible substitutions and keeps the generated prompts well-formed.
> > >
> > > ## Evaluation description and motivation
> > >
> > > > *Line 311: It is unclear what zero-shot lesion segmentation is intended to evaluate. Could you clarify what type of generalization the held-out pathology datasets are designed to test? Are we testing for generalization to unseen diseases or same diseases but unseen populations or unseen tasks, or something else?*
> > >
> > > We now clarify that in this experiment, we evaluate generalization to new pathology datasets. In particular, three of the segmentation classes used in this experiment (resection cavities, hemorrhages, and MS lesions) were completely excluded during the training process and are used to evaluate zero-shot segmentation performance on new pathology types.
> > >
> > > > *Section 4.2. What is specialist network referring to? Can you provide a citation here?*
> > >
> > > We now more explicitly define the specialist models in the main text. The specialist models used in the ablations (Section 4.2) train the VoxelPrompt vision backbone (without the language conditioning) individually on single tasks. This is designed to test whether a *single* multi-task VoxelPrompt model can match the performance of *a collection* of task-specific models.
> > >
> > > > *Could the authors clarify how the evaluations in Section 4 validate the use cases presented in Figure 1? At present, it is not clear whether VoxelPrompt successfully accomplishes its intended use case.*
> > >
> > > We should have been more clear in our text. Since there are no benchmark datasets for the quantitative, multi-step radiology workflows we execute, our primary experiments in Section 4 focused on measuring performance via standard subtasks, which also facilitated comparison against widely-used standard (e.g., SynthSeg) and VLM (e.g., BiomedParseV2, SAT) based segmentation baselines.
> > >
> > > The quantitative or morphometric region-of-interest processing tasks shown in Figure 1 involve an intermediate segmentation output (e.g., "compute the lesion growth" entails segmenting the lesion). Once the region is correctly segmented, downstream calculations and code executions are  correctly predicted, and subsequently executed, by VoxelPrompt. To evaluate this, we have now added an experiment to Section 4.1 reporting accuracy across a subset of full end-to-end workflows.
> > >
> > > ## Baselines
> > >
> > > > *Insufficient baselines. For certain evaluations, the authors compare against only a single baseline which is insufficient in evaluating how the proposed approach compares to the existing literature.*
> > >
> > > Most of the tasks we evaluate do not have established or broadly adopted baselines. Where robust and accurate external baselines exist, we compare against them: SynthSeg for modality-agnostic whole-brain segmentation, MoME for generalizable lesion segmentation, and BiomedParse-V2 and SAT for text-guided volumetric segmentation. For tasks without suitable external methods, we develop, train, and include specialist models  that use the VoxelPrompt vision backbone trained individually for single tasks, providing an architecture-matched reference.
> > >
> > > In the revised Section 4.1, we now add the widely-used SAMSEG method (an optimization-based framework) and SAT (a VLM) to the whole-brain segmentation and longitudinal effect-size estimation experiments. We also explicitly evaluate full multi-step workflows against baselines, as described in our earlier response.
> > >
> > > We emphasize that the aim of the work is not to surpass every task-specific model, but to demonstrate that a single unified system can match specialist performance while enabling integrated, end-to-end analyses, which existing models cannot perform.

---

> > > > ### Author Response · Authors · 2025-11-25
> > > > **Response to Reviewer DgCn (part 4 of 4)**
> > > >
> > > > > *Fig 3D: Why not compare performance against other VLMs? My intuition says that other VLMs will achieve a similar runtime improvement when compared against FreeSurfer.*
> > > >
> > > > To our knowledge, SAT is the only existing VLM capable of whole-brain 3D neuroanatomical structural segmentation. Its training distribution, however, is substantially narrower than the clinical variability represented in our data. In the revision, we evaluate SAT on the subset of structures overlapping its training domain, and report these results in Section 4.1 and Figure 3, where SAT achieves only a mean $12.4 \pm 25.0$% Dice accuracy.
> > > >
> > > > The runtime comparison in the original Figure 3 was not intended as a central claim. We therefore de-emphasize it and instead expand the evaluation to incorporate the above baselines. Importantly, models such as SynthSeg, SAMSEG, and SAT do not compute longitudinal change metrics. For these methods, we report segmentation accuracy and derive downstream measures manually, whereas VoxelPrompt produces them directly within a single predicted end-to-end workflow.
> > > >
> > > > > *Fig 3E: Same comment here, why did you only choose to benchmark against SynthSeg? Why not compare against other segmentation or VLM models?*
> > > >
> > > > For whole-brain anatomical segmentation, we benchmark against SynthSeg because it is the only state-of-the-art 3D learning-based method that robustly handles the full range of modalities and MRI acquisition parameters represented in our dataset. Unfortunately, besides SAT, existing 3D VLM models are not designed to tackle this task. To expand this analysis, we have expanded the comparison to include the widely-used optimization-based method SAMSEG, which we evaluate on the anatomical structures it supports, as well as the 3D VLM, SAT, as described in the response above. These results are now reported in the revised Section 4.1 and Figure 3, where VoxelPrompt outperforms SAMSEG and SAT by $5.1 \pm 6.1$% and $70.8 \pm 25.7$% Dice, respectively.
> > > >
> > > > > *In line 198, how do you guarantee that the feedback procedure does not result in an infinite loop?*
> > > >
> > > > During training, the model executes ground-truth instructions rather than the predicted outputs. At inference, the agent generates the next step in the sequence based on the learned distribution of valid task workflows. We have not observed looping in practice, likely because the tasks are supervised and well-structured, though we do implement a standard maximum-token limit as a safeguard.
> > > >
> > > > > *How well does this approach generalize to new tasks, especially more complex multi-step tasks? It seems like the training procedure is quite specific to simple tasks that are provided during training. Is the dataset generation approach scalable?*
> > > >
> > > > We appreciate the reviewer's question. We clarify that the primary aim of this work is not open-ended agent generalization to arbitrary unseen tasks, but to introduce the first framework for language-guided end-to-end workflow generation for complex medical imaging tasks. This capability is essential for supporting realistic, multi-step radiological workflows. We have emphasized this aim throughout the revised manuscript.
> > > >
> > > > At evaluation, if given an out-of-domain prompt, the output typically maps to the closest familiar task structure, which is expected given the limited task training scope, and aligns with our intended use of VoxelPrompt as an accurate, neuroimaging-specialized system, rather than an open-set coding agent.
> > > >
> > > > Regarding scalability: in clinical imaging domains, tasks must be specified explicitly because real workflows require precise, context-specific quantitative objectives that cannot be inferred reliably from general-purpose models, particularly in 3D where labeled data are scarce. Our task-generation strategy described above is easily extensible, where new capabilities can be added  by expanding the task library or  training data.
> > > >
> > > > > *What is the effect of synthetic lesion training? Can you do an ablation to show that the proposed approach is effective?*
> > > >
> > > > Thank you for the helpful suggestion. In the revised manuscript, we now have ablation results in a new Figure 5 showing that incorporating synthetic lesion training yields a $27.7$% Dice improvement in generalization to new pathology datasets.

---

### Official Review · Reviewer_dWyV · 2025-10-29

**Soundness:** 2
**Presentation:** 3
**Contribution:** 2
**Rating:** 4
**Confidence:** 5

**Summary:**

This paper presents VoxelPrompt, a system that combines a language model agent with a jointly-trained vision network to perform complex neuroimaging analysis tasks.

**Strengths:**

The joint training of language and vision components for code-based workflow generation is interesting. The code generation approach provides transparency and interpretability compared to black-box vision-language models, which is important for clinical applications.

**Weaknesses:**

1. "End-to-end" is claimed throughout, but most quantitative evaluations are on segmentation subtasks, not complete clinical workflows.
2. Training the language model from scratch on synthetic template-based prompts is a critical limitation explicitly acknowledged: "limits their utility when given entirely unseen prompts". The system can only use predefined library functions, limiting true flexibility.
3. Missing quantitative evaluation of the complex multi-step workflows shown in Figure 1. And small sample sizes for some tasks (e.g., n=12 subjects for vascular territory classification).
4. There are some Baseline Comparison Concerns. Different prompts for different baselines (shown in Table on p.19) is unclear if BiomedParse v2 and SAT received optimal prompting.
5. Fine-tuning a pretrained language model (even small ones like CodeLlama, StarCoder) would likely improve prompt generalization significantly. The choice to train from scratch seems inefficient and limits performance.
6. Some notation inconsistencies (e.g., E used for both encoder output and feature encodings).
7. The longitudinal FreeSurfer comparison is somewhat unfair. FreeSurfer performs full cortical reconstruction, not just segmentation.

**Questions:**

1. What percentage of queries result in code execution errors?
2. How does the system handle code execution failures, invalid operations, or edge cases?
3. What happens when users provide prompts significantly different from training templates?
4. Can the approach extend to other body regions (chest, abdomen) or modalities without retraining from scratch?
5. For RadFM, what performance would be achieved with the full 32-layer model? Could other memory optimization strategies enable fair comparison?

---

> ### Author Response · Authors · 2025-11-25
> **Response to Reviewer dWyV (part 1 of 2)**
>
> Thank you for the detailed feedback and constructive suggestions regarding our experiments and claims. We have made substantial edits to the manuscript in response to these comments, with updated text appearing in blue.
>
> > *"End-to-end" is claimed throughout, but most quantitative evaluations are on segmentation subtasks, not complete clinical workflows ... Missing quantitative evaluation of the complex multi-step workflows shown in Figure 1.*
>
> This should have been clearer in our text; thank you for raising it. As there are no benchmark datasets for the quantitative, multi-step radiology workflows we execute, our primary quantitative experiments focused on measuring performance via standard subtasks, which facilitated comparison against widely-used standard (e.g., SynthSeg) and VLM (e.g., BiomedParseV2, SAT) based segmentation baselines.
>
> That said, we have now added an experiment in Section 4.1 reporting accuracy across a subset of full end-to-end workflows, which include measurements of (1) volumes of anatomical regions and pathologies, (2) ventricular asymmetry, (3) resection cavity and meningioma dimensions, and (4) mean CT density of hemorrhage lesions.
>
> Additionally, we clarify that our pathology characterization experiments, presented in Appendix D, represent *classification*-related end-to-end workflows, such as identifying the neuroanatomical location of a lesion. We have also added Appendix B.1, which explicitly enumerates and details the various quantitative task categories used in our experiments. We have revised the text to make these aspects clearer.
>
> > *Training the language model from scratch on synthetic template-based prompts is a critical limitation explicitly acknowledged: "limits their utility when given entirely unseen prompts". The system can only use predefined library functions, limiting true flexibility. ... Fine-tuning a pretrained language model (even small ones like CodeLlama, StarCoder) would likely improve prompt generalization significantly. The choice to train from scratch seems inefficient and limits performance.*
>
>
> We completely agree that we do not tackle truly open-set coding capabilities. However, we respectfully clarify that this is not VoxelPrompt's goal. We chose to train a small language model from scratch for several reasons, including:
>
> First, VoxelPrompt's target domain is much narrower than that of general-purpose code generation. Quantitative region-of-interest processing, image characterization, and measurements require a much smaller set of atomic Python functions, rather than the open-ended functionality required for general software engineering. This reduces the need for high-capacity language models in this context.
>
> Second, to have high enough capacity for open-set coding, models such as StarCoder and CodeLlama have 3B and 7B parameters, respectively, at their smallest.  VoxelPrompt requires the language model to be trained jointly with a *volumetric* convolutional 3D encoder-decoder. As 3D convolutional network passes have a substantial memory footprint (often up to 40GB or more -- see Figure 4B), we trained a 70M parameter language model from scratch to work within our limited computational resources of a single A100 GPU. Training a much smaller language model from scratch enables us to maintain stable bidirectional coordination between the language and vision subnetwork, while remaining within hardware constraints.
>
> We agree that broad prompt generalization is an important goal for future universal, generalist approaches. However, VoxelPrompt's goal is to instead introduce a technical framework capable of a broad range of quantitative workflows. Such a system requires (i) language-driven instruction generation, (ii) coupled processing with a vision and segmentation network, (iii) support for arbitrary numbers and types of 3D inputs, and (iv) execution of complete analysis pipelines. Our experiments are designed to validate these core capabilities rather than evaluate open-domain code generalization. Since this distinction was not adequately made in the original paper, we have revised the abstract, introduction, and discussion for clarity; thank you for raising these points.
>
> > *What happens when users provide prompts significantly different from training templates?*
>
> When prompted using out-of-domain prompts, the language model currently produces code or language corresponding to a familiar task seen in training. As described in the question above, we have revised the paper to incorporate these discussions.

---

> > ### Author Response · Authors · 2025-11-25
> > **Response to Reviewer dWyV (part 2 of 2)**
> >
> > > *Small sample sizes for some tasks (e.g., n=12 subjects for vascular territory classification).*
> >
> > We agree that some of the task-specific datasets are small. This is unavoidable as *annotated* volumetric data in public medical imaging datasets is typically much more limited than in natural vision. Most datasets have only tens or hundreds of annotated scans.
> >
> > Specifically, for the pathology characterization example raised by the reviewer, these annotated datasets did not exist. We manually annotated hundreds of images over several weeks in consultation with neuroimaging experts for use in these experiments, which limited the sample size available for these subtask experiments. We have expanded the limitations section to acknowledge this.
> >
> >
> >
> > > *Different prompts for different baselines (shown in Table on p.19) is unclear if BiomedParse v2 and SAT received optimal prompting.*
> >
> > We should have conveyed this more clearly. To avoid disadvantaging baselines, each baseline was evaluated using a wide range of prompts on validation data, and we selected the best-performing prompt. We reviewed each baseline's training data and documentation to make sure our prompts for BiomedParseV2 and SAT were appropriate. For added transparency, Appendix C.2 now details the prompting strategy for each baseline.
> >
> > > *Some notation inconsistencies (e.g., E used for both encoder output and feature encodings).*
> >
> > Thank you for pointing this out. We have now reviewed, improved, and standardized notation throughout the paper. Also, we now include Appendix Table 1 that maps each core variable to its description.
> >
> > > *The longitudinal FreeSurfer comparison is somewhat unfair. FreeSurfer performs full cortical reconstruction, not just segmentation.*
> >
> > We agree that this is a nuanced discussion. Longitudinal FreeSurfer (FS-long) algorithmically *requires* the full cortical reconstruction of both the input volumes and the estimated within-subject atlas to estimate the final segmentations. It is plausible that one could ablate FS-long to disable the cortical reconstruction, but that might negatively impact its intermediate registration and segmentation steps.
> >
> > For fairness, as suggested, we have now removed the emphasis in the text on the runtime gains w.r.t.~FS-long and focus instead on matching it in effect size. We now also include results from two additional baselines (SynthSeg and SAMSEG) in the longitudinal analysis.
> >
> > > *What percentage of queries result in code execution errors? How does the system handle code execution failures, invalid operations, or edge cases?*
> >
> > In training, we execute only ground-truth code, so execution errors do not occur. During evaluation, any invalid code produced by the model is executed in the environment, and any resulting exceptions are propagated to the user.
> >
> > > *For RadFM, what performance would be achieved with the full 32-layer model? Could other memory optimization strategies enable fair comparison?*
> >
> > We believe there is a miscommunication on our part. We *did* evaluate the full, original 32-layer RadFM model, but found that it did not generalize to any of our evaluation tasks without finetuning. We now better clarify this in the paper.
> >
> > We therefore made a best-faith attempt at finetuning it on our dataset within our limited computational resources (a single 80GB A100 GPU) and spent multiple weeks on mixed precision memory optimizations and gradient checkpointing. Despite these efforts, we found that the original RadFM model was not trainable within single-GPU memory, regardless of optimizations. The original RadFM model was trained on 32 80GB A100 GPUs and already used memory optimization strategies such as automatic mixed precision and gradient checkpointing. We therefore opted to use the first eight transformer blocks of RadFM as a pretrained initialization instead, as this configuration was the largest trainable variant under these constraints.
> >
> > > *Can the approach extend to other body regions (chest, abdomen) or modalities without retraining from scratch?*
> >
> > The VoxelPrompt training strategy is designed to be agnostic to the anatomical region of interest. For example, the instruction structure, code execution loop, and multimodal vision processing pipeline do not assume brain-specific priors. We therefore expect the system to be able to extend to other regions and modalities when trained with domain-specific data. We now further emphasize this in the introduction and discussion sections.

---

### Official Review · Reviewer_dG9U · 2025-10-31

**Soundness:** 1
**Presentation:** 1
**Contribution:** 2
**Rating:** 2
**Confidence:** 4

**Summary:**

This paper propose a code agent framework to assist radiologist of free-form radiological tasks. Experimental reuslts show the current system seems work well in some author defined tasks.

**Strengths:**

The paper is well-written in structure and readers are easy to follow.

**Weaknesses:**

1. An Inefficient and "Weak" Agent: The paper's most significant limitation is that its agent is trained from scratch on a curated, domain-specific dataset. This results in an agent that is, by definition, "weaker" and less capable in its reasoning, language understanding, and code-generation abilities than any modern, general-purpose foundation model (such as the Gemini or GPT series). The field has largely demonstrated that the emergent reasoning and planning capabilities of large-scale models are a prerequisite for robust agentic behavior.

2. Ignores the Superior LGM-as-Agent Paradigm: As you noted, the tasks described (e.g., "calculate progressive signal reduction") are complex, multi-step analytical workflows. The current, established approach for this is to use a powerful, pre-trained foundation model as a central "agent" or "orchestrator." This agent then intelligently calls upon a suite of specialized tools—which could include the VoxelPrompt vision network itself—via APIs. The paper's design, which builds a weak, custom agent instead of leveraging a powerful general one, seems to solve the wrong problem.

3. Lack of Verifiable Generalizability: The "from-scratch" approach makes the agent "brittle." Because the vision and language models are co-trained on this specific neurological dataset, the agent's "intelligence" is inextricably tied to this single domain. There is no evidence it could generalize to any other task (e.g., analyzing chest X-rays, a slightly different MRI protocol). This lack of zero-shot capability makes it impossible to verify the agent's generalizability, as its performance is completely coupled with its training data.

**Questions:**

See Weaknesses for details.

---

> ### Author Response · Authors · 2025-11-25
> **Response to Reviewer dG9U (part 1 of 2)**
>
> Thank you for your feedback. It has been helpful to better contextualize our work in the revision. We believe that there is a central miscommunication regarding our goals and contributions, and we appreciate the opportunity to clarify them below.
>
> > *An Inefficient and "Weak" Agent: The paper's most significant limitation is that its agent is trained from scratch on a curated, domain-specific dataset. This results in an agent that is, by definition, "weaker" and less capable in its reasoning, language understanding, and code-generation abilities than any modern, general-purpose foundation model (such as the Gemini or GPT series). The field has largely demonstrated that the emergent reasoning and planning capabilities of large-scale models are a prerequisite for robust agentic behavior.*
>
> VoxelPrompt was developed as a dedicated system for 3D neuroimage analysis, *not* a general-purpose AI agent.  VoxelPrompt's use-cases involve text-prompted region-of-interest segmentation, quantification, and morphometrics common in neuroimage analysis workflows. These workflows use a curated, domain-relevant set of atomic python functions to execute (e.g., counting the number of segmentation voxels).
> Our primary aim is to support diverse medical imaging aims by building a flexible, resolution-agnostic, multi-input, and text-prompted vision network for volumetric 3D vision -- this does not yet exist. Constructing one requires *joint* vision-language training such that the language network can generate embeddings that can control the flexible vision network.
>
> To our knowledge, there is currently **no existing system** that plans and executes arbitrary workflows on volumetric medical images. If the reviewer is aware of such work, we would be happy to cite and compare against it. VoxelPrompt represents a first step towards achieving this goal. While it is *intentionally constrained* in its free-form text-generation capabilities, it achieves state-of-the-art zero-shot segmentation on new and unseen pathologies, can perform volumetric image characterization, and can perform text-prompted image segmentation on volumes with higher flexibility than any existing model.
>
> Given our focus on building the first controllable, instruction-conditioned 3D vision model, we allocate capacity to the vision network rather than to a large general-purpose LLM. Additionally, general purpose LLMs are far harder to constrain to the precise, domain-specific behaviors required for our complex 3D neuroimage workflows.
>
> We agree that our use of the word "agent" can be confusing and overlap with tool-using frameworks. For clarity, in the revision, we have removed the word "agent" and changed the title and writing to better align with our contributions -- thank you for raising this point. The title is now "VoxelPrompt: A Vision System for Language-Prompted Medical Image Analysis" and will be reflected in the camera ready version as OpenReview only allows title edits after the discussion period.
>
>
> > *Ignores the Superior LGM-as-Agent Paradigm: As you noted, the tasks described (e.g., "calculate progressive signal reduction") are complex, multi-step analytical workflows. The current, established approach for this is to use a powerful, pre-trained foundation model as a central "agent" or "orchestrator." This agent then intelligently calls upon a suite of specialized tools—which could include the VoxelPrompt vision network itself—via APIs.*
>
> We respectfully clarify that, while individual task-specific 3D neuroimaging tools exist, a sufficiently flexible "*suite of specialized tools*" for real-world neuroimage analysis **does not exist**, and addressing this gap is a central motivation for our work. For example, no existing segmentation tool supports the description-based targeting of volumetric ROIs illustrated in Figure 3A. Rather than requiring many narrowly tailored tools callable through an API, VoxelPrompt provides a single 3D vision network conditioned by language embeddings to support this broad landscape of tasks. What the reviewer suggests, having an LLM call the VoxelPrompt vision network, would require VoxelPrompt to exist in the first place.
>
> To train a large, capable, and flexible vision network within academic hardware constraints, we traded off some language flexibility by training a language network from scratch. However, even with this trade-off, VoxelPrompt still outperforms current SOTA biomedical Vision-Language Models (Figure 3C). We would be happy to benchmark against any biomedical tool-using LLM for segmentation, morphometrics, and quantitative analysis, if the reviewer points us to them.
>
> We agree with the reviewer’s suggestion that integrating a general-purpose tool-using LLM with VoxelPrompt is a highly compelling direction for future extensions, and we have expanded the discussion of future work accordingly.

---

> > ### Author Response · Authors · 2025-11-25
> > **Response to Reviewer dG9U (part 2 of 2)**
> >
> > > *The paper's design, which builds a weak, custom agent instead of leveraging a powerful general one, seems to solve the wrong problem*
> >
> > We respectfully disagree. Instead of the wrong problem, VoxelPrompt tackles a **different** problem from what the reviewer is envisioning. As clarified above, we do not seek to use an agent (like GPT or Gemini) that calls on tools for image analysis, because these flexible tools do not yet exist (e.g., that can do the kind of ROI targeting shown in Figure 3A). VoxelPrompt seeks to capture this multitude of required (and currently unavailable) tools into a single network, and our method uses language model embeddings to do so.
> >
> > We understand that this can be easily misunderstood by readers, as our language model is not a free-form agent. We have therefore removed the word "agent" from the paper to provide clarity and to better contextualize our work.
> >
> > > *Lack of Verifiable Generalizability: The "from-scratch" approach makes the agent "brittle." Because the vision and language models are co-trained on this specific neurological dataset, the agent's "intelligence" is inextricably tied to this single domain. There is no evidence it could generalize to any other task (e.g., analyzing chest X-rays, a slightly different MRI protocol). This lack of zero-shot capability makes it impossible to verify the agent's generalizability, as its performance is completely coupled with its training data.*
> >
> > There is a critical miscommunication. Our goal is to build a neuroimage analysis framework. Our primary experiments (Fig 3A, 3B, 3C) include **zero-shot applications** of our model to lesions and neuropathologies it has never been exposed to. For example, VoxelPrompt has not been trained on anything resembling the resection cavities in EPISURG. In these experiments, we find that VoxelPrompt broadly outperforms foundation models both specifically trained for brains (MoME) and for general-purpose medical image segmentation (BiomedParseV2, SAT). Further, while both of these general-purpose foundation model baselines (BiomedParseV2 and SAT) use text conditioning from pretrained large-scale language models, they are outperformed by VoxelPrompt, which uses a much simpler from-scratch language model training.
> >
> > Regarding the specific examples the reviewer raised:
> > - "*analyzing chest X-rays*": Throughout the paper, we focus on neuroimage analysis, and do not intend to generalize beyond neuroimaging. While our paper does provide a general framework that could be extended to other anatomical regions given adequate training data, that is not our focus.
> > - "*a slightly different MRI protocol*": As clarified above, our experiments in Fig. 3B and 3C involve zero-shot generalization to completely held out datasets with different MRI protocols (e.g., different clinical resolutions), age distributions (e.g., pediatric presentations of MS), pathology types (e.g., resections), and more.
> >
> > We have now further emphasized and clarified this zero-shot verifiable generalizability across the paper in the revision.
> >
> >
> >
> > > *The paper is well-written in structure and readers are easy to follow.*
> >
> > Thank you for highlighting our writing and structure being easy to follow for readers. Given this assessment, we are confused by the reviewer's presentation rating for the paper being the lowest possible ("1: poor"). We would appreciate any clarification on this so that we can improve our presentation as needed.
> >
> >
> > ### Closing note:
> >
> > We appreciate the reviewer's feedback, as it enables us to identify key areas of miscommunication and addressing them has improved our manuscript greatly. We appreciate you raising these points, and we look forward to discussing any remaining aspects.

---

### Official Review · Reviewer_RYFR · 2025-11-04

**Soundness:** 3
**Presentation:** 2
**Contribution:** 4
**Rating:** 8
**Confidence:** 5

**Summary:**

The paper introduces VoxelPrompt, an end-to-end system that combines language-driven code generation with a unified vision model. The method combines a language model and a cnn by allowing the language model to augment the cnn with code, and thus perform unconstrained analysis of the input.

**Strengths:**

The proposed method stands out for its originality, offering a fundamentally new approach in both functionality and design compared to existing medical image frameworks. Its flexibility and performance are highly compelling.

* The innovative use of code as an output of a language model, enabling segmentation to serve its true role as an intermediate step toward downstream clinical or analytical objectives.
* A thorough and convincing evaluation that clearly demonstrates the method’s effectiveness.
* The joint training of a language model and a segmentation model from scratch, a novel and effective strategy.
* The ability to process scans at native resolution is useful and non-trivial with cnns.
* The used datasets are interesting, however underspecified. In particular the q&a pairs, would, if released, represent a contribution to the community.

**Weaknesses:**

The primary limitation of the paper lies in the presentation of the proposed method. Its exact capabilities and constraints are not clearly defined. Crucial details regarding the architecture, datasets, and evaluation are relegated to the appendix, forcing the reader to consult supplementary material to grasp the approach.

* The use of the term “zero-shot” is misleading, as the model has been exposed to the same ROIs during training; the evaluation therefore reflects domain transfer rather than true zero-shot performance.

* The evaluation section omits essential information about the baseline specialist models, preventing a sound assessment of comparative performance. Similarly, the available tools and interfaces accessible to the language model are not specified.

* The term “agent” is used without sufficient justification.

* Key procedural details are missing regarding the generation of Q&A pairs: it is unclear whether templates are used, what their variation is, and whether the language model generalizes beyond them.

* An ablation study examining the language model’s ability to correctly identify and interpret the intended task is also absent, limiting interpretability of the reported results.

* Lastly, the paper lacks important information on the tools available for the agent to use -- i.e. which functions can be called when and in what order.

**Questions:**

* Can the proposed network be zero-shot adapted to arbitrary tasks, or is adaptation limited to tasks represented in the training dataset?
* How many distinct regions of interest (ROIs) are included in the dataset?
* In this context, does “zero-shot” refer to ROIs unseen during training?
* What are the specialist models referenced in Figure 4A, and do they represent current state-of-the-art baselines?
* Do all evaluated tasks involve segmentation, or are other modalities or objectives included?
* Will the datasets used in the study, particularly the Q&A pairs, be publicly released?
* What is the solution space available to the language model? Which functions can be called when and in what order?

---

> ### Author Response · Authors · 2025-11-25
> **Response to Reviewer RYFR (part 1 of 2)**
>
> Thank you for your positive feedback and constructive suggestions regarding our technical descriptions and presentation. We have now updated the manuscript to integrate your feedback into the submission, with the revised text in blue.
>
> ## Technical details
>
> > *The paper's exact capabilities and constraints are not clearly defined. Crucial details regarding the architecture, datasets, and evaluation are relegated to the appendix, forcing the reader to consult supplementary material to grasp the approach.*
>
> We agree and have now substantially revised the manuscript to clarify the method's capabilities, assumptions, and limitations. Specifically, we have moved key architectural details into the main text, and expanded Sections 3 and 4 to provide clarifications on training data generation and evaluation methodology. We discuss these revisions further below.
>
> > *The evaluation section omits essential information about the baseline specialist models, preventing a sound assessment of comparative performance. What are the specialist models referenced in Figure 4A, and do they represent current state-of-the-art baselines?*
>
> We now explicitly define the specialist models in the revised Section 4.2. The specialist models used in the ablations train the VoxelPrompt vision backbone (without the language conditioning) individually on single tasks. This is designed to test whether our *single* multi-task VoxelPrompt model can match the performance of *a collection* of task-specific models.
>
> Alongside these ablations, we also compare against external state-of-the-art baselines for whole-brain segmentation ([SynthSeg](https://www.sciencedirect.com/science/article/pii/S1361841523000506)), and recent foundation models for brain lesions ([MoME](https://ieeexplore.ieee.org/document/10879789/)) and text-prompted segmentation ([BiomedParseV2](https://github.com/microsoft/BiomedParse/tree/v2), ([SAT](https://www.nature.com/articles/s41746-025-01964-w)). These represent the current state-of-the-art for these segmentation problems, which we match or exceed in segmentation accuracy.
>
> > *Key procedural details are missing regarding the generation of Q&A pairs: it is unclear whether templates are used, what their variation is, and whether the language model generalizes beyond them.*
>
> We now expand the description of our template construction process in Section 3.2 to better specify how we vary prompt semantics and sample diverse descriptions of target regions of interest. We also include a template illustration and examples of sampled prompts in Appendix Figure 6.
>
> The revised manuscript is now clear about the fact that we do not expect the language model to generalize to out-of-domain prompts. In such scenarios, it maps to the closest familiar task text output. This aligns with our intended use of VoxelPrompt as a domain-specialized system, rather than an open-set coding agent.  We have better emphasized this aim throughout the revised manuscript. While large pretrained language models and automated task synthesis may broaden task generalization and further improve results in complementary future work, our focus is on developing a technical foundation for language-conditioned generation of clinical image analysis workflows.
>
> > *Lastly, the paper lacks important information on the tools available for the agent to use -- i.e. which functions can be called when and in what order. What is the solution space available to the language model?*
>
> The language model predicts code to solve in-domain tasks using built-in Python functions, volumetric and morphological imaging functions from the *voxel* package (included as supplementary material), as well as `encode` and `segment` functions that execute the vision model. These are functions made available in the Python execution environment. We have now made this clear in Sections 3.1 and A.2.

---

> ### Author Response · Authors · 2025-11-25
> **Response to Reviewer RYFR (part 2 of 2)**
>
> ## Experiment and framing clarifications
>
> > *Do all evaluated tasks involve segmentation, or are other modalities or objectives included?*
>
> Our quantitative or morphometric region-of-interest (ROI) processing tasks involve intermediate segmentation. However, our qualitative pathology characterization tasks, evaluated in Appendix D, represent *classification*-related end-to-end workflows that do not directly involve segmentations, such as identifying the neuroanatomical location of a lesion.
>
> For quantitative ROI processing tasks, once the region is correctly segmented, downstream calculations and code executions are predicted, and subsequently executed, by VoxelPrompt. Since there are no benchmark datasets for the quantitative, multi-step radiology workflows we execute, our primary quantitative experiments focused on measuring performance on standard subtasks, which also facilitated comparison against widely-used standard (e.g., SynthSeg) and VLM (e.g., BiomedParseV2, SAT) based segmentation baselines.
>
> To better evaluate more complex tasks, we have added an experiment in Section 4.1 reporting accuracy across a subset of full end-to-end workflows, which include computation of measurements of (1) volumes of anatomical regions and pathologies, (2) ventricular asymmetry, (3) resection cavity and meningioma dimensions, and (4) mean CT density of hemorrhage lesions.
>
> We have also added Appendix B.1, which explicitly enumerates and details the various task categories used in our experiments. We have revised the text to make these details more clear.
>
>
> > *Can the proposed network be zero-shot adapted to arbitrary tasks, or is adaptation limited to tasks represented in the training dataset?*
>
> The proposed network can generalize to new tasks that are semantically similar to the training tasks. For example, even though VoxelPrompt has never been trained on resection cavities, it can accurately segment them with a prompt that contains lesion descriptors, such as "*segment the hypointense lesion*". However, it cannot be zero-shot applied to entirely new, arbitrary tasks such as registration or denoising. We now clarify this in the discussion. Thank you for raising this point.
>
> > *An ablation study examining the language model’s ability to correctly identify and interpret the intended task is also absent, limiting interpretability of the reported results.*
>
> We employed multiple levels of analysis to explore the ability of VoxelPrompt to interpret and solve the intended task. As discussed above, we do not expect or attempt to generalize to tasks that require out-of-domain processing pipelines. We did evaluate the diverse ways in which language can prompt context-specific targets (Figure 3A). Across our analyses, see Section 4.1, we find that prompt-selected segmentation and end-to-end analysis of in-domain ROIs matches or exceeds single-tasks baselines. We find no outliers in which VoxelPrompt computes a segmentation corresponding to a label that does not match the prompted ROI. Further, across our experiments, we find no examples where VoxelPrompt fails to generate the correct code (as determined by the task-specific ground truth).
>
> > *The use of the term "zero-shot" is misleading, as the model has been exposed to the same ROIs during training; the evaluation therefore reflects domain transfer rather than true zero-shot performance.*
>
> We believe there might be a miscommunication. Three of the segmentation classes used in this experiment (resection cavities, hemorrhages, and MS lesions) were completely excluded during the training process. These targets were segmented using the prompts "segment the hypointense lesion" for resection cavities, "segment the hyperdense lesions" for brain hemorrhages, and "segment hyperintensities" for the MS lesions, as detailed in Appendix C.2. We therefore believe it is fair to consider this zero-shot transfer to disease type. We have now made this more explicit in the paper.
>
> > *The term “agent” is used without sufficient justification.*
>
> We agree that the term "agent" may have caused confusion. In VoxelPrompt, the language model interprets the user prompt and generates the end-to-end workflow for neuroimage analysis. While this resembles modern LLM-based agents, our focus is fundamentally on developing the joint vision–language framework needed for language-conditioned volumetric image processing, rather than on building a general-purpose reasoning or planning agent. For clarity, we have removed the emphasis on the term "agent" in the paper.
>
> > *How many distinct regions of interest (ROIs) are included in the dataset?*
>
> We have 185 unique anatomical ROIs for anatomical classes and 14 unique labels for pathology classes.
>
> > *Will the datasets used in the study, particularly the Q&A pairs, be publicly released?*
>
> We will release an end-to-end neuroimaging Q&A pair benchmark dataset with the paper.

---

> > ### Comment · Reviewer_RYFR · 2025-11-27
> >
> > Thank you for the response. Your changes to the manuscript substantially improved the clarity of the paper, which is appreciated. I fully endorse the acceptance of this paper and withhold my score at 8.

---

### Author Response · Authors · 2025-11-25
**Overall Response**

We thank all reviewers for their thoughtful and constructive feedback. Based on their suggestions, we have substantially revised  the paper to improve its technical clarity and methodological presentation and to provide a more comprehensive quantitative evaluation. We have also reframed our contributions and limitations following reviewer feedback. All revised text appears in blue.

We appreciate that reviewers highlighted the utility and novelty of the proposed framework as a medical imaging system (`RYFR`, `dWyV`, `DgCn`), the relevance of joint vision–language training and interpretable code-based workflows (`RYFR`, `dWyV`, `DgCn`), the clinical applicability of the task space evaluated (`RYFR`, `DgCn`), and the clarity of presentation (`RYFR`, `dG9U`).

Additionally, in response to reviewer suggestions and questions, the revised manuscript includes major updates including, but not limited to, the below:
- We have added new experiments that benchmark VoxelPrompt's accuracy on end-to-end workflows (Fig. 3F) and an ablation demonstrating the benefits of our lesion synthesis strategy (Fig. 5).
- New baselines including popular iterative and vision-language based frameworks have been added to existing experiments on longitudinal effect size estimation (Fig. 3D) and anatomical brain segmentation (Fig. 3E).
- We have overhauled the Methods section for clarity and have expanded technical descriptions of key architectural details, training task and prompt generation, lower-level mechanisms (e.g., cross-volume interaction), and all other relevant details.
- A clearer statement of scope: the goal of this work is to introduce and evaluate the first jointly trained system capable of handling diverse, end-to-end 3D volumetric clinical imaging workflows, and not open-domain agent generalization for tool usage. We have removed the focus on the agent terminology to avoid this confusion.

We address each specific point in the individual responses below. Again, we appreciate the feedback and believe it has improved the clarity and completeness of the work substantially. We gladly welcome any further discussion and suggestions.

---

### Author Response · Authors · 2025-12-02
**Comment to the New AC**

We thank the new AC for their efforts under these uncharted circumstances. The initial reviews, the rebuttal, and the revisions made to the submission are summarized below.

Reviewers `RYFR`, `dWyV`, and `DgCn`  acknowledged the overall technical novelty and utility of our work, but had concerns regarding the initial submission's presentation clarity, framing, experimental details, and our choice of baselines. In the discussion period, we significantly revised the manuscript to incorporate and address each point of reviewer feedback. These edits included clarifications of key technical components, expanded methodological and experimental details, narrowing of claims, and the addition of several new experiments and baselines suggested by the reviewers. Additionally, `dG9U`'s primary concern pertained to our model's lack of "agentic" capabilities, which we believe stems from a miscommunication of scope that we have now accounted for throughout the revision.

To summarize our interactions with each reviewer:

**(1)** Reviewer `RYFR` requested clearer definitions of the method's capabilities and constraints, alongside further technical details. We substantially revised Sections 3-4 to update key architectural, training, and evaluation details, and we expanded our discussion of specialist baselines, state-of-the-art baselines, prompt templates, and available functions. The reviewer also requested clearer experimental framing, which we addressed through both new end-to-end workflow results and new text. We additionally clarified the intended semantic generalization regime and removed emphasis on the "agent" term.

**(2)** Reviewer `dG9U`'s primary concern did not pertain to any technical or experimental issues that were specific to our paper; they instead wanted us to execute an entirely different approach. The review assessed our system as a free-form reasoning agent (comparable to GPT/Gemini) that calls on pre-existing tools. We believed the source of this misunderstanding may be our use of the "agent" and "free-form" terms in the initial submission. Our revised paper removed these terms and clarified that our contribution is instead a novel text-prompted, multi-volume 3D vision model to support common *real-world* medical imaging workflows related to segmentation, morphometrics, and characterization (and not free-form analysis). Our revision further clarified that the specialized tools the reviewer expects do not exist for neuroimaging, and our method is precisely designed to fill this gap.

**(3)** Reviewer `dWyV` noted that our claims could be better supported by new experiments and raised several points of experimental and technical clarification regarding training on templates, sample sizes, the prompting and characterization of baselines, and training from scratch. In the revision, we added new experiments to address each of the reviewer's comments and expanded our descriptions of the scope of our prompt space and intended use cases, baseline usage, and all other methodological and experimental details. We believe we carefully addressed all expressed concerns and, in the process, substantially strengthened the paper.

**(4)** Reviewer `DgCn` primarily raised several low-level technical questions spanning notation, architecture, task generation, and evaluation design. We addressed each point and made corresponding revisions throughout the paper to integrate their feedback, including clearer notation, an expanded architectural description, a detailed account of task construction and templates, clearer descriptions of evaluation strategies, and new appendices documenting tasks. The reviewer also noted that some of our experiments could have more baselines and ablations. To remedy this concern, we added new end-to-end workflow experiments and the reviewer's requested additional baselines and ablations to the manuscript. We believe we strengthened the technical clarity and depth of the paper substantially by incorporating and addressing each point raised by the reviewer.

Post-rebuttal, Reviewer `RYFR` maintained their initial score of eight and Reviewers `dG9U`, `dWyV`, `DgCn` did not have the chance to respond to our rebuttal before the author-reviewer discussion period was closed. We are grateful for the reviewers' feedback as it has substantially improved our paper.

---

### Meta-Review · Area_Chair_TAAp · 2026-01-02

**Summary:**

This paper presents VoxelPrompt, a language-guided programming system for 3D medical image analysis that generates executable code from natural language queries. The approach enables a unified framework for segmentation, localization, and quantitative measurements through joint training of a language model and a segmentation network.

The submission received divergent reviews with scores of 8, 4, 2, and 2. While one reviewer (RYFR) strongly endorsed the paper's novelty and methodology, the remaining three reviewers raised substantial concerns regarding technical clarity, the mismatch between claimed contributions and experimental validation, limited generalization capabilities, and insufficient baseline comparisons. Only RYFR participated in the post-rebuttal discussion, maintaining their positive score, while the other three reviewers did not respond.

**Reviewer Concerns:**

Concerns Adequately Addressed: The authors made commendable efforts to improve the manuscript. Technical presentation was substantially revised with expanded architecture descriptions, a unified symbol table (Appendix Table 1), and clearer documentation of training procedures. RYFR explicitly acknowledged these improvements. The authors appropriately recalibrated their claims by removing the term "agent" and revising the title. Additional baselines (SAMSEG, SAT) and a new ablation study on lesion synthesis (Figure 5) were included. End-to-end workflow experiments were added in Section 4.1 to address the evaluation gap.

Outstanding Concerns:

The core tension between claimed end-to-end workflow capability and subtask-focused evaluation remains inadequately resolved. While the authors added workflow experiments in Section 4.1, the sample sizes are small and the benchmarks are self-constructed rather than established clinical standards.

Reviewer dWyV's concern about training from scratch versus fine-tuning pretrained language models was not experimentally addressed. The authors provide justification based on hardware constraints, but this remains a methodological limitation that could affect prompt generalization.

Reviewer dG9U's concerns, while framed around a different problem scope than the authors intended, raise valid points about the system's brittleness and domain-specific limitations. The authors acknowledge the system cannot generalize to out-of-domain prompts, which limits practical utility.

The baseline comparison issues raised by multiple reviewers were only partially addressed. Comparisons against SAT and SAMSEG were added, but the evaluation still relies heavily on specialist ablations rather than comprehensive external benchmarks.

**Reviewer Scores:**

RYFR (original: 8): Maintained at 8.

dWyV (original: 4): Would likely remain at 4. Core concerns about end-to-end evaluation and prompt generalization were partially but not fully addressed.

DgCn (original: 2): Would likely remain at 2. Many technical clarity issues were addressed, but concerns about evaluation design motivation and baseline sufficiency remain partially unresolved.

dG9U (original: 2): Would likely remain at 2. The fundamental methodological disagreement about training from scratch versus leveraging pretrained LLMs represents a philosophical difference that cannot be resolved through revision. The authors raised valid concerns about the quality of this review, which I have considered in my evaluation.

While one reviewer was enthusiastic, the consensus among the remaining reviewers identified fundamental concerns about evaluation scope and methodology that were not fully resolved by the rebuttal.

---

### Decision · Program_Chairs · 2026-01-26

Reject